# PhenoSV: interpretable phenotype-aware model for the prioritization of genes affected by structural variants

Zhuoran Xu[1,2,3], Quan Li [4], Luigi Marchionni[3] & Kai Wang [2,5] ✉

Structural variants (SVs) represent a major source of genetic variation associated with phenotypic diversity and disease susceptibility. While long-read sequencing can discover over 20,000 SVs per human genome, interpreting their functional consequences remains challenging. Existing methods for identifying disease-related SVs focus on deletion/duplication only and cannot prioritize individual genes affected by SVs, especially for noncoding SVs. Here, we introduce PhenoSV, a phenotype-aware machine-learning model that interprets all major types of SVs and genes affected. PhenoSV segments and annotates SVs with diverse genomic features and employs a transformer-based architecture to predict their impacts under a multiple-instance learning framework. With phenotype information, PhenoSV further utilizes gene-phenotype associations to prioritize phenotype-related SVs. Evaluation on extensive human SV datasets covering all SV types demonstrates PhenoSV's superior performance over competing methods. Applications in diseases suggest that PhenoSV can determine disease-related genes from SVs. A web server and a command-line tool for PhenoSV are available at https://phenosv.wglab.org.

A major source of genomic variation between individuals comes from structural variants (SVs). SVs are typically defined as genomic rearrangements spanning over 50 base pairs (bp), including deletions, duplications, insertions, inversions, translocations, and complex rearrangements[1–4]. Due to their large size, SVs can affect a higher fraction of the human genome, producing more pronounced molecular and phenotypic consequences than single nucleotide variations (SNV)[5–8]. SVs can directly alter gene dosage when they encompass the entire or partial gene coding regions. SVs can also indirectly influence gene expressions when coding regions of the gene are outside of the SV-affected area, by impinging on their transcriptional machinery, regulatory elements, or genome 3D structures[9,10]. Although most SVs are benign, resulting in no or only adaptive phenotypic effects[11], many

of them can lead to phenotypic variations including various human diseases[10,12,13], such as Parkinson's disease[14], Crohn's disease[15], AIDS[16], neurodevelopmental disorders[17], and cancers[18–20]. Somatic SVs also play significant roles in human diseases: according to the ICGC/TCGA Pan-Cancer Analysis of Whole Genomes (PCAWG) Project, more than 60% of cancer driver mutations are related with SVs[21]. Earlier generations of genomic techniques, such as SNP microarrays and short-read sequencing, can only generate a small number of SVs reliably, typically deletions and duplications[4]. With the increased popularity of long-read sequencing (LRS), the 2022 "Methods of the Year" featured by the journal *Nature Methods*[22], over 20,000 SVs per human genome can be detected[4,23]. Thus, identifying and prioritizing pathogenic or functionally important SVs relating to an individual's phenotypes from tens

[1]Graduate Group in Genomics and Computational Biology, University of Pennsylvania Perelman School of Medicine, Philadelphia, PA 19104, USA. [2]Raymond G. Perelman Center for Cellular and Molecular Therapeutics, Children's Hospital of Philadelphia, Philadelphia, PA 19104, USA. [3]Department of Pathology and Laboratory Medicine, Weill Cornell Medicine, New York, NY 10065, USA. [4]Princess Margaret Cancer Centre, University Health Network, University of Toronto, Toronto, ON M5G2C1, Canada. [5]Department of Pathology and Laboratory Medicine, University of Pennsylvania, Philadelphia, PA 19104, USA. ✉e-mail: wangk@chop.edu

of thousands of candidate SVs, including small noncoding SVs and translocations/inversions, becomes an essential and challenging task now.

Many computational tools have been developed in recent years for SV annotation and interpretation, and they can be broadly classified as aggregation-based, rule-based, and machine-learning-based methods. Aggregation-based methods, such as SVScore[24], use summary statistics of a single type of SNV pathogenicity scores for all positions inside the SV to derive SV pathogenicity scores. However, using one single site-based feature may not be able to account for diverse types of functional impacts for SVs that cover a large collection of sites. Rule-based methods, such as AnnotSV[25] and SvAnna[26], annotate and score SVs based on a series of criteria, with the ability to relate functional impacts of SVs to the gene level. Yet, AnnotSV relies heavily on previously observed SVs, with potential limitations in interpreting novel SVs. SvAnna similarly depends on existing genome annotations, but it has simplified rules to recognize SVs with more complex pathogenic mechanisms, such as the impacts of SVs on distal genes, and cannot make predictions when patient phenotype information is not available. Machine-learning-based methods, on the other hand, are powerful in making in silico predictions by considering various types of genetic features. For instance, Sharo et al. developed a random-forest-based method StrVCTVRE[27] to distinguish pathogenic coding SVs of deletions and duplications that overlap exons from benign ones. By integrating a diverse set of genomic features, StrVCTVRE exhibited promising results and outperformed previous aggregation-based and rule-based methods. Similarly, SVPath[28] was proposed to predict the pathogenicity of human exonic SVs that are deletions, duplications, or insertions, based on gradient boosting decision tree. Since 98% of human SVs are noncoding[29], CADD-SV[30] and X-CNV[31] are designed to make genome-wide predictions of the functional effects of SVs using similar approaches, but they have decreased performance for noncoding SVs and cannot make predictions for inversions and translocations. SVFX[32] is another standalone framework designed to train disease-specific models based on random forest using SVs from disease groups and control groups, thus enabling quantification of SV pathogenicity. However, it cannot be directly applied to novel variants. Recently, Althagafi et al. proposed DeepSVP[33] to prioritize disease-related SVs given phenotype information. Their results suggest the value of utilizing genotype-phenotype relationships in SV interpretations. However, DeepSVP can only make gene-level predictions based on genes directly covered/impacted by SVs and has limitations in inferring the impacts of SVs on distal genes. In summary, while several approaches are available to predict pathogenic SVs, especially machine-learning based ones that integrate multiple sources of information, there are limitations in the interpretability of the models, the types of SVs that can be analyzed, and the ability to prioritize phenotype-relevant genes that explain the functional relevance of predicted pathogenic SVs.

Predicting the functional consequences of SVs, especially noncoding SVs and some less-studied SV types (translocations, inversions and complex rearrangements), remains challenging for several reasons. First, it can be difficult to determine the disease-related genes from a set of genes that are all affected by SVs directly or indirectly. Second, although several large-scale experimental mapping studies have been conducted to determine genomic functional elements[34–36], annotations in noncoding portions of the human genome still lagged behind those in coding regions, due to the many different ways in which how noncoding regions influence genome function. Third, the regulatory circuits in the human genome are complex, with multiple regulatory modules working together to fine-tune gene expression levels. Diverse forms of genome rearrangements can disrupt 3D chromatin organizations and even impact genes at considerable distances, further complicating SV interpretations[9,13]. Lastly, interpreting inversions, insertions, and translocations is particularly challenging

due to limitations in existing labeled datasets that are built primarily based on short-read sequencing and chromosomal microarrays. Currently, no machine-learning-based methods can assess the functional impacts of all major types of SVs on individual genes, both within and outside of genomic regions covered by SVs.

To overcome these challenges, we developed PhenoSV, a machine-learning-based method to predict the functional consequences of coding and noncoding SVs on clinical or cellular phenotypes. To address the first challenge, we dissect the impacts of SVs on individual genes by employing a multiple-instance learning (MIL) framework, which enables us to make inferences on both the SV level and the gene level. For the second challenge, SVs are annotated using a diverse set of genetic features in a per-segment fashion to address distinct mechanisms of how specific coding and noncoding genome regions impact genome function. For the third challenge, PhenoSV adopts a transformer-based architecture with masked multi-head attention to model indirect and long-range regulatory effects of noncoding SVs on genes. For the last challenge, we use different forms of deletions and duplications that PhenoSV was trained with to approximate the impacts of inversions, insertions, and translocations. PhenoSV was extensively tested in a large collection of human SV datasets. Our results demonstrate that PhenoSV can accurately predict the pathogenicity of both coding and noncoding SVs of all SV types, with the ability to capture distinct predictive features within the same model. When phenotype information is available (such as in the form of Human Phenotype Ontology), PhenoSV can further utilize gene-phenotype associations (such as those documented in databases or predicted by other algorithms) to improve SV prioritizations. Applications of PhenoSV in a diverse range of SV datasets on human diseases and phenotypes suggest that PhenoSV can identify pathogenic SVs responsible for different phenotypes, as well as critically important genes directly or indirectly affected by SVs, to enable the prioritization of candidate SVs from a large candidate list and to facilitate the interpretation of disease association studies.

## Results

### PhenoSV overview

The overall workflow of PhenoSV for deletions and duplications is illustrated in Fig. 1. We first segment the genome region of interest that is potentially affected by a given SV into a sequence of genome segments and annotate the SV in a per-segment fashion (Fig. 1a). For a coding SV, we mainly consider its direct effects on genes that are fully or partially encompassed by the SV. Hence, the genome segment sequence comprises the covered protein-coding genes and intergenic noncoding regions. For a noncoding SV, we consider its indirect effects on genes within a given distance (e.g., 1Mbp) or topological associating domain (TAD)[37–39]. The segment sequence for this SV then includes candidate protein-coding genes affected, intergenic noncoding regions, and the noncoding SV itself. We add zero-padding segments at the front and end of every genome segment sequence. All genome segments are then annotated by 238 features derived from 64 annotation types of 6 functional categories, including deleteriousness scores, epigenetic activities, disease constraints, genome annotations, evolutionary constraints, and SV types (Supplementary Data 1). The annotated SV is fed into a transformer-based architecture[40] with masked multi-head attention (MHA) mechanisms, enforcing separate heads to model the direct and indirect effects on genes (Fig. 1b). All gene-level embeddings are then aggregated into the SV-level embedding through a max-pooling layer, followed by a classifier and a calibrator to predict general pathogenicity of overall SV (PhenoSV scores, $p_{sv}$) and individual genes (PhenoSV gene scores, $p_{sv-gene}$) from SV- and gene-level embeddings, respectively. The impacts of insertions, inversions, and translocations can be correspondingly approximated using basic forms of deletions and duplications (see "Methods" and Fig. 2). When prior knowledge of the patient's phenotypes is available,

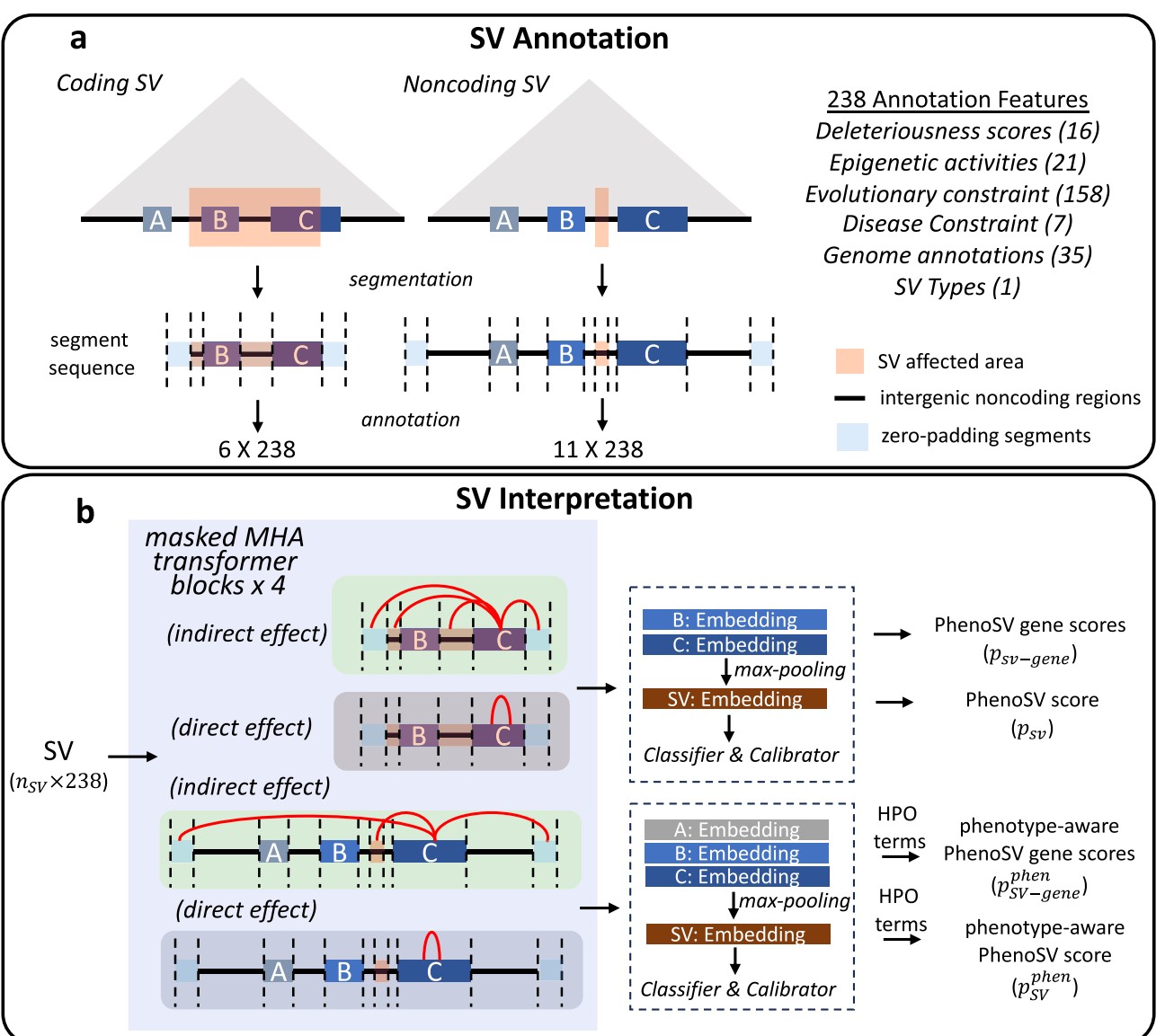

**Fig. 1 | PhenoSV workflow. a** SV annotation. A coding SV that is a deletion or a duplication, fully containing gene B and partially encompassing gene C, is segmented into a sequence of six genome segments, including two affected genes, two intergenic noncoding regions, and two zero-padding segments. A noncoding SV that is a deletion or a duplication can potentially affect gene A, B, and C based on distance or TAD annotations (triangle shaded area). The genomic segment sequence has three candidate target genes, five intergenic noncoding regions, a noncoding SV region, and two zero-padding segments. **b** SV interpretation.

Annotated SV with the shape of 6 ×238 or 11 ×238 from (**a**) is fed into PhenoSV architecture. Each MHA (multi-head attention) block has two types of attention heads to model indirect and direct effects on genes. The pathogenicity for overall SV (PhenoSV scores, $p_{sv}$) and individual genes (PhenoSV gene scores, $p_{sv-gene}$) can be inferred from SV-level and gene-level embeddings, respectively. Prior phenotype information (HPO terms) can be further used to infer phenotype-related pathogenicity for overall SV (phenotype-aware PhenoSV scores, $p_{sv}^{phen}$) and individual genes (phenotype-aware PhenoSV gene scores, $p_{sv-gene}^{phen}$).

we employ existing gene-phenotype scoring methods, such as Phen2Gene[41], to derive gene-phenotype or SV-phenotype association scores. These scores are then used to refine the general pathogenicity scores into phenotype-specific pathogenicity scores on the SV level (phenotype-aware PhenoSV scores, $p_{sv}^{phen}$) and the gene level (phenotype-aware PhenoSV gene scores, $p_{sv-gene}^{phen}$).

**PhenoSV training and testing**

PhenoSV was trained and extensively tested on a large collection of curated human SV datasets (see "Methods" and Supplementary Data 2). Coding SVs of deletions and duplications from ClinVar[42] and a matched noncoding SV dataset of deletions and duplications were combined and split into training, validation, and hold-out test sets by chromosomes (Table S6). SVs of chromosomes 11, 12, and 13 were used for the validation set, and those of chromosomes 14, 15, and 16 were used for the hold-out test set. The rest of the SVs were used for training. The training dataset contains 14,622 SVs, with 14,292 being coding SVs (6609 pathogenic and 7683 benign) and 330 being noncoding SVs (165 pathogenic and 165 benign). The validation dataset contains 2182 SVs (2136 coding SVs: 990 pathogenic and 1146 benign; 46 noncoding SVs: 23 pathogenic and 23 benign), and the hold-out test dataset contains 2673 SVs (2559 coding SVs: 1385 pathogenic and 1174 benign; 114 noncoding SVs: 57 pathogenic and 57 benign). Coding SVs of deletions and duplications from DECIPHER[43] and SvAnna[26] that do not have extensive overlaps with SVs from ClinVar were used as independent test sets, further splitting into a small SV dataset (50 bp–100 kbp, 383 pathogenic and 366 benign SVs) and a large SV dataset (100 kbp–1 Mbp, 1208 pathogenic and 801 benign SVs). SVs of insertions (175 pathogenic, 175 benign), inversion (20 pathogenic, 20 benign), and translocations (68 pathogenic, 38 benign) were also

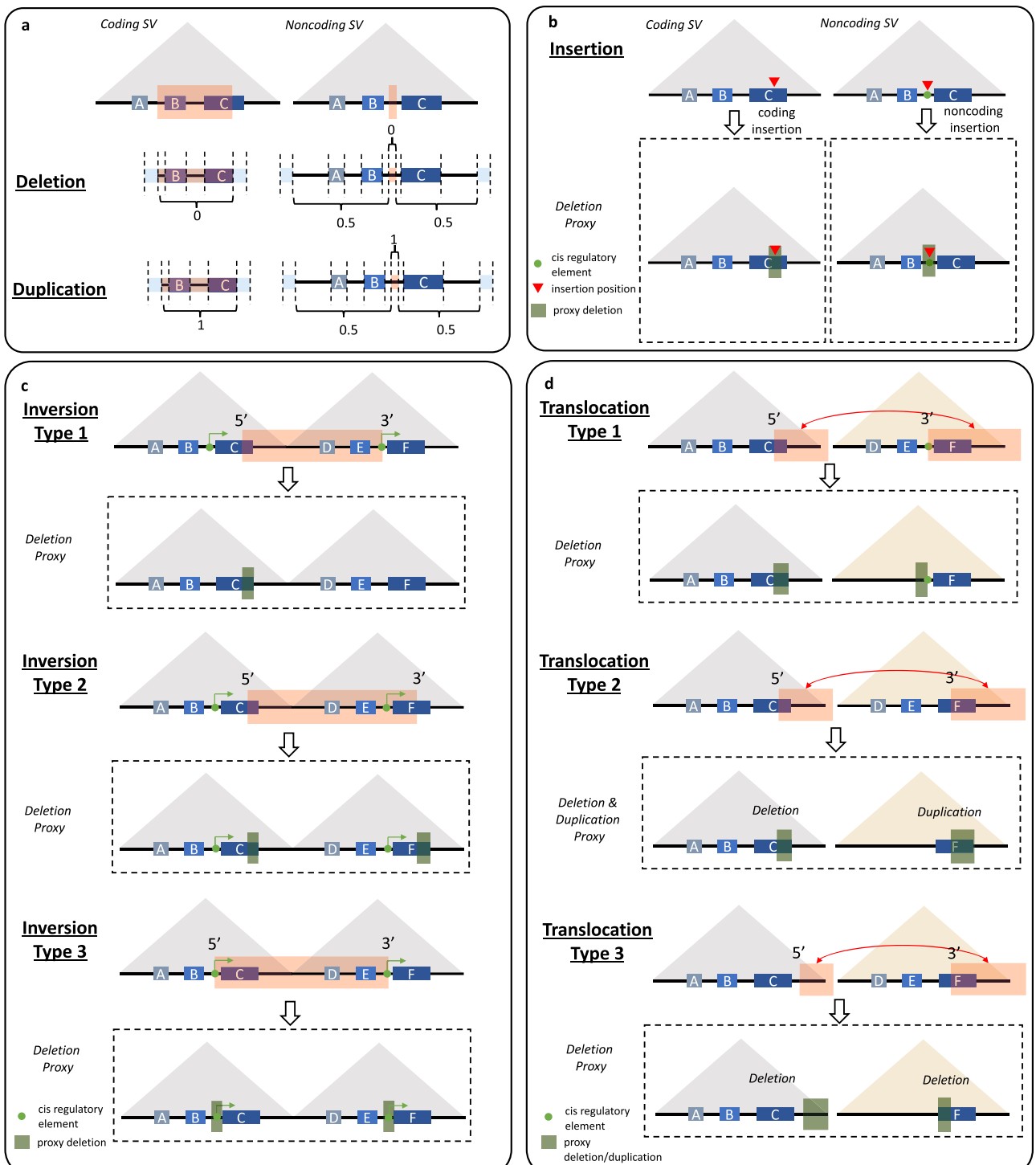

**Fig. 2 | PhenoSV workflow to assess the impacts of all major types of SVs.**
**a** PhenoSV dissects a given SV that is deletion or duplications into a sequence of genomic segments. The last feature dimensions of genomic segments within the SV region are encoded as 0 for deletions, 1 for duplications, and 0.5 for those outside the SV region. **b** An insertion is treated as a 100 bp deletion. Displayed are a coding insertion directly disrupting gene C, and a noncoding insertion disrupting a regulatory element that indirectly affects gene A, B, and C. **c** Inversion type 1 (one breakpoint within genes): we only consider the impacts of its 5′ breakpoint as a deletion truncating the gene C. Inversion type 2 (two breakpoints within genes): we consider impacts of both 5′ breakpoint and 3′ breakpoint as two deletions truncating the gene C and the gene F. Inversion type 3 (no breakpoint within genes): we consider the impacts of both 5′ breakpoint and 3′ breakpoint as a 100 bp deletion centered at the 5′ breakpoint indirectly affecting genes A, B, and C, and a 100 bp

deletion centered at the 3′ breakpoint indirectly affecting genes D, E, and F.
**d** Displayed are translocations swapping two genome segments. Impacts of both 5′ and 3′ breakpoints of translocations are considered. Translocation type 1 (gene truncation): the 5′ breakpoint is treated as a deletion truncating the 3′ side of gene C. The 3′ breakpoint is treated as a deletion losing a segment of intergenic region that can indirectly affect the gene F. Translocation type 2 (gene fusion): this resulted fusion gene C−F is treated as a deletion truncating the 3′ side gene C (decreased copy number of the segment) and a duplication of the 5′ side of gene F (increased copy number of the segment). Translocation type 3 (gene truncation): the 5′ breakpoint is treated as a deletion truncating the 3′ side genome region of the breakpoint and indirectly affect genes A, B, and C. The 3′ breakpoint is treated as a deletion truncating the 5′ of the gene F.

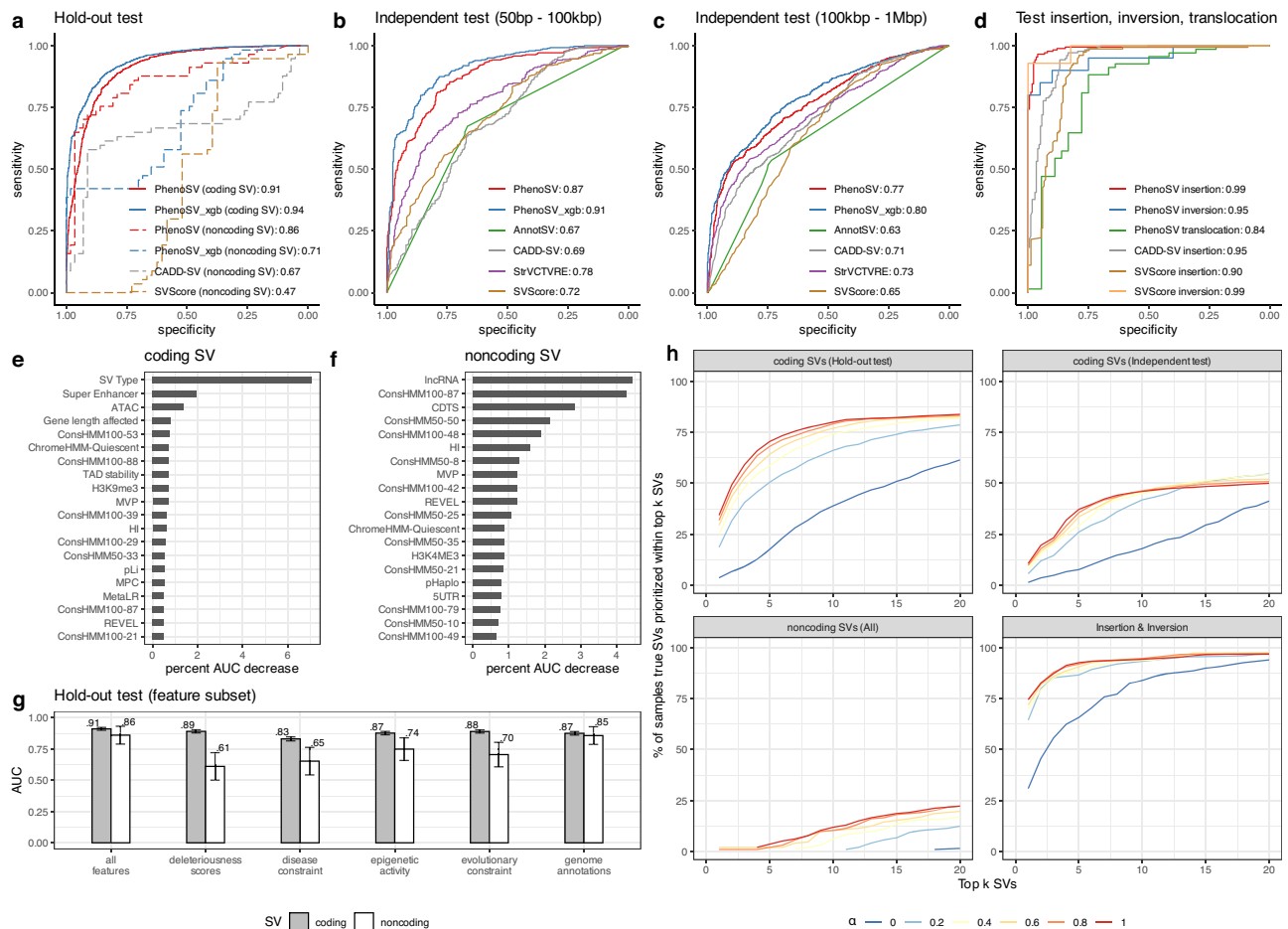

**Fig. 3 | Evaluation of the performance of PhenoSV and feature importance for coding and noncoding SVs. a** Model AUCs in the hold-out test dataset for coding SVs ($n = 1385$ pathogenic and $n = 1174$ benign SVs, solid lines) and noncoding SVs ($n = 57$ pathogenic and $n = 57$ benign SVs, dashed lines). **b, c** Model AUCs in the independent test datasets of small coding SVs ($n = 383$ pathogenic and $n = 366$ benign SVs) with sizes ranging from 50 bp to 100 kbp and large coding SVs ($n = 1208$ pathogenic and $n = 801$ benign SVs) with sizes ranging from 100 kbp to 1 Mbp. **d** Model AUCs in the test datasets of insertions ($n = 175$ pathogenic SVs and $n = 175$ benign SVs), inversions ($n = 20$ pathogenic SVs and $n = 20$ benign SVs), and translocations ($n = 68$ pathogenic fusion transcripts and $n = 38$ benign fusion transcripts). **e, f** PhenoSV feature importance measured by percent AUC decrease in

the hold-out test dataset for coding and noncoding SVs. **g** PhenoSV AUCs in the hold-out test dataset for coding and noncoding SVs trained with all 238 features or only a subset of features belonging to the same category (*x* axis). Error bars represent 95% CI. **h** PhenoSV performance in prioritizing phenotype-related SVs. Displayed are percentage of samples that the true disease-related SV is prioritized (*y*-axis) within top k (*x*-axis) out of about 19,000 SVs. $\alpha$ controls for the contributions of phenotype information in prioritization. True disease-related SVs are from coding SVs in hold-out test set (top left panel), coding SVs in independent test set (top right panel), all noncoding SVs (bottom left panel), and SVs of insertion and inversion (bottom right panel). All SVs in (**a**–**c**) and (**e**–**g**) are deletions or duplications. Source data are provided as a Source Data file.

compiled from multiple resources as test datasets. Pathogenic SVs that have phenotype information were collected to evaluate PhenoSV's performance in prioritizing disease-related SVs, including 1007 coding SVs from the hold-out test dataset, 494 coding SVs from the independent test dataset, all 193 noncoding SVs compiled, and 149 SVs of insertions and inversions. We used three human disease datasets of congenital abnormalities[44], autism[45], and epilepsy[46] to evaluate PhenoSV performance on the gene level. We further curated a dataset containing SVs affecting similar genome regions but linking to either inherited diseases[47] or cancers[48] to illustrate how PhenoSV helps identify critical genes associated with distinct phenotypes.

## PhenoSV accurately predicts pathogenicity of both coding SVs and noncoding SVs

To evaluate the performance of PhenoSV in predicting overall SV pathogenicity without phenotype information, we compared PhenoSV (using $p_{sv}$) to several representative methods belonging to three categories mentioned earlier. These include: (1) PhenoSV-XGBoost, which is trained with a traditional machine learning method of XGBoost[49] using the same set of features as PhenoSV; (2) SVScore, an

aggregation-based method that uses summary statistics of a single type of SNV pathogenicity scores; (3) AnnotSV, a rule-based method that relies heavily on previously observed SVs; (4) StrVCTVRE, a traditional machine learning-based method for only coding SVs of deletions and duplications; and (5) CADD-SV, a traditional machine learning-based method for both coding and noncoding-SVs applicable for deletions, duplications, and insertions. As not all methods can produce scores with natural choices of thresholds that distinguish between pathogenic and benign SVs, we used the area under the receiver-operating characteristic curve (AUC) as a performance metric to compare different methods. We also reported accuracy, sensitivity, and specificity of PhenoSV in Table S1, and area under the precision-recall curves (auPRC) in Fig. S4.

Since PhenoSV was trained on deletions and duplications, we first evaluate the model performance in the test datasets containing only deletions and duplications. For small coding SVs in the independent test dataset (383 pathogenic and 366 benign SVs, 50 bp–100 kbp, Fig. 3b), PhenoSV and PhenoSV-XGBoost that use the same 238 genetic features achieved AUC of 0.87 (95% CI: 0.85–0.90) and 0.91 (95% CI: 0.89–0.93), respectively, much improved than SVScore (AUC: 0.72,

95% CI: 0.68–0.76), AnnotSV (AUC: 0.67, 95% CI: 0.64–0.71), CADD-SV (AUC: 0.69, 95% CI: 0.65–0.73), and StrVCTVRE (AUC: 0.78, 95% CI: 0.75–0.81). For large coding SVs in the independent test dataset (1208 pathogenic and 801 benign SVs, 100 kbp–1 Mbp, Fig. 3c), the AUC is 0.77 (95% CI: 0.75–0.79) for PhenoSV and 0.80 (95% CI: 0.77–0.82) for PhenoSV-XGBoost, still outperforming SVScore (AUC: 0.65, 95% CI: 0.63–0.68), AnnotSV (AUC: 0.63, 95% CI: 0.61–0.76), CADD-SV (AUC: 0.71, 95% CI: 0.69–0.74), and StrVCTVRE (AUC: 0.73, 95% CI: 0.71–0.85). These results indicate that machine learning-based methods can generate more precise predictions by efficiently incorporating various genomic features and our integrated feature set is more informative to aid in SV pathogenicity predictions (Fig. 3b, c). Because StrVCTVRE cannot make genome-wide predictions and AnnotSV treats ClinVar SVs as known, we can only compare PhenoSV with PhenoSV-XGBoost, SVScore, and CADD-SV for noncoding SVs in the hold-out test dataset (Coding SVs: 1385 pathogenic and 1174 benign SVs; Noncoding SVs: 57 pathogenic and 57 benign SVs). As shown in Fig. 3a, PhenoSV (AUC: 0.86, 95% CI: 0.79–0.93) significantly outperformed PhenoSV-XGBoost (AUC: 0.71, 95% CI: 0.61–0.80), SVScore (AUC: 0.47, 95% CI: 0.35–0.60) and CADD-SV (AUC: 0.67, 95% CI: 0.56–0.77) in predicting the pathogenicity of noncoding SVs, suggesting the importance of incorporating contextual information of candidate target genes for noncoding SVs. Using the same set of features, PhenoSV-XGBoost consistently outperformed PhenoSV for coding SVs (deletions and duplications, Fig. 3a–c), suggesting the advantages of traditional machine learning methods for tabular data due to lower model complexities compared with deep learning methods[50]. However, the extra complexities brought by SV segmentation and the transformer-based architecture dramatically helped PhenoSV achieve a higher AUC for noncoding SVs than PhenoSV-XGBoost (Fig. 3a), indicating a more critical role of the regulatory information between noncoding segments and genes in noncoding SVs than coding SVs. More importantly, PhenoSV can generate interpretable results by making predictions on the gene level, which is a common limitation for traditional machine-learning methods. Notably, we observed lower AUCs of large SVs than small SVs in the independent test sets for all models except CADD-SV (Fig. 3b, c). This performance decrease for larger SVs could be attributed to potential ascertainment biases arising from disparities in the techniques used for SV detection (see Supplementary Materials). Additionally, we evaluated PhenoSV's performance in interpreting SVs (2034 pathogenic and 1934 benign SVs) located on sex chromosomes. As shown in Fig. S6, PhenoSV generalizes well for SVs on sex chromosomes, achieving an AUC of 0.94 (95% CI: 0.93–0.95).

Due to the presence of purifying selection pressure, allele frequency is expected to inversely correlate with the functional significance of mutations[51]. To avoid potential ascertainment biases during the selection of training dataset, we deliberately excluded allele frequency from the input feature set during the training of PhenoSV. However, we can now employ allele frequency as a performance metric to evaluate whether predicted PhenoSV scores exhibited the anticipated negative correlation with allele frequency in the hold-out test set. The estimation of allele frequencies was carried out based on gnomAD-SV database[52]. As expected, PhenoSV scores are negatively correlated with SV allele frequency, where rarer SVs are more likely to be pathogenic (Spearman's rho = −0.19, $p$ value < 0.0001).

## PhenoSV accurately predicts pathogenicity of insertion, inversions, and translocations

We then tested whether PhenoSV could be used for insertions, inversions, and translocations, whose functional impacts are approximated by different forms of deletions and duplications (see "Methods" and Fig. 2). Here, only SVScore and CADD-SV can be used as the competing methods for insertions because other methods are either not applicable for these SV types or treat labeled SVs from ClinVar as known. Figure 3d shows that PhenoSV achieved almost perfect performance for insertions (175 pathogenic and 175 benign SVs, AUC: 0.99, 95% CI: 0.98–1.00), outperforming SVScore (AUC: 0.90, 95% CI: 0.87–0.94) and CADD-SV (AUC: 0.95, 95% CI: 0.93–0.97). The AUC of PhenoSV for inversions is 0.95 (20 pathogenic and 20 benign SVs, 95% CI: 0.88–1.00), where SVScore achieved AUC of 0.99 (95% CI: 0.96–1.00). Although most translocations, especially those resulting in fusion genes, are pathogenic and have been viewed as one of the hallmarks of cancer, recent studies have demonstrated the existence of benign chimeric RNAs in non-disease tissues[53]. We thus used a test dataset composed of common chimeric RNAs in non-disease tissues and cancers to evaluate the performance of PhenoSV in predicting the pathogenicity of translocations. Figure 3d shows PhenoSV can accurately distinguish benign chimeric RNAs from pathogenic ones (68 pathogenic and 38 benign SVs), achieving an AUC of 0.84 (95% CI: 0.74–0.93).

## Distinct feature sets contribute to pathogenicity predictions of coding SVs and noncoding SVs

The rationale for integrating various feature categories is to comprehensively capture genetic variations and jointly make robust predictions. To increase the interpretability of the machine learning models, we investigated the contributions of each feature category in predicting the pathogenicity of coding and noncoding SVs. Besides the PhenoSV model trained with all 238 features, we additionally trained five models using different subsets of features, in which only features of the same category and SV type feature were used. For both coding and noncoding SVs, PhenoSV performs the best when using all feature categories jointly, where deleteriousness scores contribute the most for coding SV predictions, and genome annotations play an essential role in noncoding SV predictions (Fig. 3e). We then examined individual feature importance by permuting the input matrix on each feature dimension and measuring the percent AUC decrease in the hold-out test dataset. As shown in Fig. 3f, g, coding and noncoding SV predictions are driven by distinct feature sets. Specifically, SV type is the most important feature, followed by super-enhancer annotations for coding SV predictions. In contrast, the top two for noncoding SV predictions are lncRNA and genome conservation state annotation. To investigate how PhenoSV identifies predictive features driving pathogenicity calls of distinct SVs, we applied it to two novel SVs discovered recently and are not in its training dataset or existing databases. We first examined a novel 10,678 bp in-frame deletion (chr15:48452562-48463240, GRCh38) identified by Elgaz et al.[54]. This deletion affects exons 42–45 of the *FBN1* gene and results in neonatal Marfan syndrome (nMFS), a severe form of Marfan syndrome (MFS, OMIM: 154700). Notably, the deletion is distinct from the *FBN1* region (exons 24–32) linked to the majority of nMFS cases. PhenoSV predicted this SV as pathogenic with a high confidence score ($p_{sv} = 0.961$). We then analyzed the key features driving this pathogenicity call based on values of input x gradient and found REVEL score, haploinsufficiency score, 20-way phyloP Score, MVP score, and MetaLR score are the top 5 most important features (Fig. S1a). We also examined a novel SVA insertion[55] in *SRCAP* exon 13 (chr16:30712369-30712370, GRCh38) causing Floating-Harbor syndrome (FLHS, OMIM: 136140), which are typically caused by truncating variants in *SRCAP* exons 33-34. PhenoSV similarly predicted this SV as pathogenic ($p_{sv} = 0.99$), where REVEL score, 20-way phyloP Score, MVP score, CCRS score, and replication timing are the top 5 most important features driving the prediction to pathogenic (Fig. S1b). Collectively, these results demonstrate the ability of PhenoSV to capture distinct predictive features from diverse genome annotations to make accurate pathogenicity predictions for both coding and noncoding SVs and highlight the potential of PhenoSV as a tool for interpreting the impacts of newly discovered SVs where existing knowledge is limited.

## PhenoSV can utilize prior phenotype information for improved SV prioritization

We then examined whether clinical phenotype information can help better prioritize disease-related SVs for a specific patient. To do so, we simulated patients' SV profiles and compared the percentage of patients whose true disease-related SVs were prioritized within the top 20 SVs (top ~0.1% of the simulated SV profile including ~19,000 SVs) using $p_{sv}^{phen}$ with different settings of $\alpha = 0, 0.2, 0.4, 0.6, 0.8$, and 1. Here, $\alpha$ controls for different degrees of phenotype dependency, from $\alpha = 0$ with no phenotype information involved to higher values of $\alpha$ with more dependence on phenotype information. Each simulated patient's SV profile contains one true disease-related pathogenic SV and ~19,000 non-overlapping control SVs as noises, of which ~8000 are rare novel SVs, including deletions, duplications, insertions, and inversions (see "Method" for details). SvAnna was applied in the same simulated SV profiles as the benchmark method (Table S4). As shown in Fig. 3h, PhenoSV generally prioritized more true disease-related SVs within top 20 when $\alpha$ is larger and when more detailed phenotype information is utilized, such as SVs in the hold-out test set from ClinVar than those in the independent test set from DECIPHER and SvAnna. Specifically, PhenoSV prioritized true disease-related coding SVs for 61.5% and 41.3% of the patients in the hold-out and independent test set, respectively, when not using any phenotype information ($\alpha = 0$). By utilizing more phenotype information and setting $\alpha = 1$, these numbers were increased to 83.8% and 50%, respectively. In comparison, SvAnna identified more true disease-related coding SVs (97.12%) in the hold-out test set, but fewer (20.64%) in the independent test set than PhenoSV ($\alpha = 1$), indicating that SvAnna places more emphasis on gene-phenotype associations to prioritize SVs while PhenoSV considers both SV features and gene-phenotype associations. Identifying noncoding pathogenic SVs can be challenging, but PhenoSV was still able to recognize true disease-related SVs for 1.55% of patients ($\alpha = 0$). By incorporating phenotype information, the precision significantly increased to 22.3%, whereas SvAnna only achieved 8.29%. In addition, PhenoSV exhibited ability to prioritize insertions and inversions with precisions being 94.0% and 96.8% for $\alpha = 0$ and $\alpha = 1$, respectively, yet the precision is 87% for SvAnna.

## PhenoSV identifies disease-related genes directly affected by coding SVs

We have demonstrated how PhenoSV can accurately predict the pathogenicity of SVs using $p_{sv}$ or $p_{sv}^{phen}$. However, since each SV can impact multiple genes, it is also essential to dissect the impacts of SVs on the gene level and determine disease-related genes in genetic studies on human diseases. To examine this, we first applied PhenoSV to 91 coding SVs from a cohort with congenital abnormalities[44]. Since each patient's HPO terms are available, we correlated $p_{sv-gene}^{phen}$ ($\alpha = 1$) with four confidence groups of disease-related genes predicted by the original study. As shown in Fig. 4a, PhenoSV produces consistent results on the gene level with the original study in general, where genes in a higher confidence group exhibit higher $p_{sv-gene}^{phen}$ scores (Spearman's rho = 0.556, $p < 0.0001$). We also noticed several outlier genes with high $p_{sv-gene}^{phen}$ scores in the non-driver gene group that are less consistent with predictions from the original study. We then investigated the SV (chr16:28473235-30186830, deletion, GRCh38) that produced the outlier gene (MAPK3) with the highest $p_{sv-gene}^{phen}$ score. Within the 57 protein-coding genes affected by this SV, two genes (SH2B1 and PRRT2) were categorized as tier 2 disease-related genes in the original study, and six genes (CLN3, TUFM, KIF22, ALDOA, TBX6, and CORO1A) were categorized as tier 3 genes. The average $p_{sv-gene}^{phen}$ score for tier 2, tier 3, and the rest of the genes are 0.52, 0.45, and 0.16, respectively. We noticed that MAPK3 was not categorized as a disease-related gene by the original study, but it had the highest $p_{sv-gene}^{phen}$ score of 0.95 by PhenoSV. Our result is supported by a previous study by Park et al., providing evidence that MAPK3 plays a potential role in the pathogenesis of autism spectrum disorder (ASD) using Drosophila models[56]. Additionally, MAPK3 is located in a 593 kb recurrent deletion region on 16p11.2, which is associated with neurodevelopmental disorders, and MAPK3 has a Simons Foundation Autism Research Initiative (SFARI)[57] gene score of 2 ("Strong Candidate"), with 11 rare variants (10 missense variants, one stop loss variant) reported in genetic studies on ASD. To elucidate how each genetic feature from an individual genome segment contributes to the pathogenic prediction of this SV, we visualized input x gradient values within each genome segment (Fig. 4b). Although segments of protein-coding genes play a major role in determining the $p_{sv}$ score as expected, noncoding intergenic segments still make contributions by capturing important features such as context-dependent tolerance score (CDTS)[58] and lncRNA annotations. Individual features, such as CADD score, haploinsufficiency score (HI), and 20-way phyloP score, are the major contributors to the general pathogenicity of MAPK3. These results further demonstrate the advantages of PhenoSV in interpreting coding SVs on the gene level.

## PhenoSV determines disease-related genes indirectly affected by noncoding SVs

To evaluate gene-level PhenoSV predictions for noncoding SVs, we applied PhenoSV to an ASD dataset with 222 noncoding SVs that disrupt cis-regulatory elements of variant-intolerant genes (CRE-SVs)[45]. In the original study, paternally inherited CRE-SVs were observed to be over-transmitted to affected offspring and not to their unaffected siblings, whereas maternally inherited CRE-SVs did not show this pattern. Consistent with findings of the original study (Fig. 4c), we found an over-transmission of paternally inherited CRE-SVs to cases (67/100; transmission rate = 67%; binomial test $p$ value = 0.0009), and maternally inherited CRE-SVs were not significantly different from the expected 50% transmission rate (47/79; transmission rate = 59%; binomial test $p$ value = 0.11). After stratifying CRE-SVs into pathogenic ($p_{sv-gene} \geq 0.5$) and benign ($p_{sv-gene} < 0.5$) groups using PhenoSV, a slightly larger effect size of over-transmission pattern was observed for paternally inherited pathogenic SVs (29/41; transmission rate = 71%; binomial test $p$ value = 0.01) than benign SVs (38/59; transmission rate = 64%; binomial test $p$ value = 0.04). Although statistical significance was not achieved due to limited sample sizes (pathogenic SVs vs. benign SVs, two-sided proportion test $p$ value = 0.656), these results suggest the values of PhenoSV in determining pathogenic genes indirectly affected by noncoding SVs. When classifying pathogenic CRE-SVs and benign CRE-SVs, different thresholds of $p_{sv-gene}$ (such as top and bottom 30% quantiles) can be used and we found that different thresholds do not influence the overall conclusion of the analysis (Supplementary Materials, Tables S7 and S8). We also applied PhenoSV to 373 noncoding SVs from an epilepsy cohort, with 150 SVs from patients and the rest 223 from controls[46]. The original study revealed that rare noncoding SVs near epilepsy genes were enriched in the patient group compared with the control group. Instead of using distance to the nearest epilepsy gene as a pathogenicity proxy, we used $p_{sv}$, $p_{sv-gene}$ of the most-affected epilepsy gene, and $p_{sv-gene}$ of the closest epilepsy gene to evaluate each SV's effects on epilepsy-related genes between patients and controls. We did not observe a significant difference in overall SV pathogenicity ($p_{sv}$) between patients and control individuals (see the left panel of Fig. 4d). Yet, we found significantly more epilepsy-related SVs in the patient group, which do not necessarily affect the nearest epilepsy genes (see the middle and the right panel of Fig. 4d), indicating $p_{sv-gene}$ of the most-affected epilepsy gene is a better proxy than $p_{sv-gene}$ of the closest epilepsy gene. Again, these results demonstrate the values of PhenoSV in determining disease-related genes from SVs, including genes beyond the genomic regions of SVs that are not the closest genes to the SVs.

To further examine what features within and outside of noncoding SV segments contribute to PhenoSV predictions, we investigated

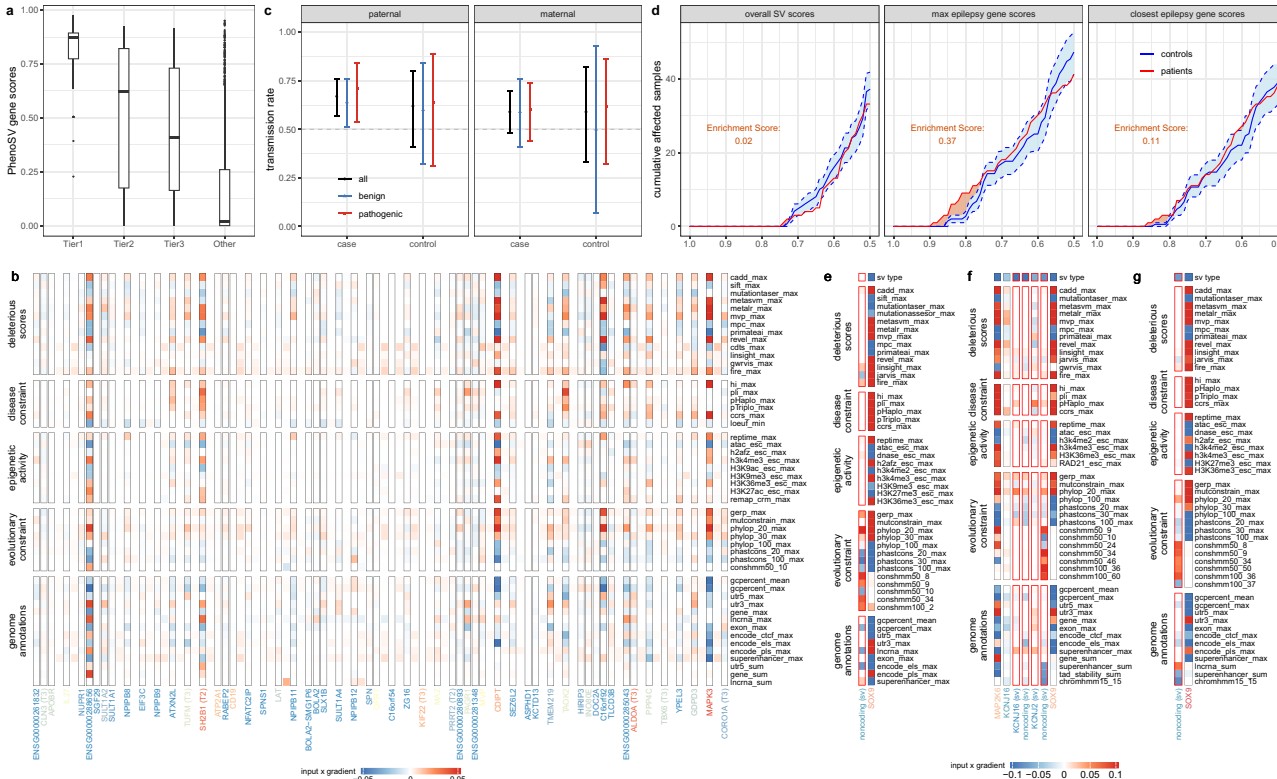

**Fig. 4 | Evaluation of the performance of PhenoSV in identifying phenotype-related genes affected by SVs. a** Boxplot of $p_{sv-gene}^{phen}$ scores (y-axis, $\alpha = 1$) in four groups of driver genes affected by 91 coding SVs predicted by the original study on congenital abnormality (x-axis). Median (center line), IQR (box limits), and outliers (points) that exceeding 1.5x IQR were shown in the boxplot. **b** Displayed are input × gradient values of top 50 important features (row) for each genome segment (column) of the SV (chr16:28473235-30186830, deletion, GRCh38), including protein-coding genes segments (black borders) and intergenic noncoding segments (no borders). Colors of column names represent values of $p_{sv-gene}^{phen}$ ranging from 0 (blue) to 1 (red) with $\alpha = 1$. **c** Transmission rate of paternal (case: $n = 100$, control: $n = 26$, left panel) and maternal (case: $n = 79$, control: $n = 17$, right panel) noncoding SVs to cases and controls with (blue and red) and without (black) being stratified by $p_{sv-gene}$. Error bars represent 95% CI of transmission rate, where the center is observed transmission rate. **d** Displayed are cumulative affected sample numbers (y-axis) with $p_{sv}$ (left panel), $p_{sv-gene}$ of the most affected epilepsy gene

(middle panel), and $p_{sv-gene}$ of the nearest epilepsy gene (right panel) larger than given thresholds (x-axis) of 150 patients and 150 controls on average. Confidence intervals of controls (blue shade area between dashed lines) are calculated by randomly sampling 150 samples from 223 controls for 100 times. Area of orange shades represent enrichment score, defined as integrated cumulative number of affected patient samples over upper bound of the 95% confidence interval of controls. **e–g** Displayed are input × gradient values of top 50 important features (row) for each genome segment of the SVs affecting *SOX9* gene (**e**: Gordon et al.[59], GRCh38, chr17:70685120-70964563, deletion; **f**: Kurth et al.[60], GRCh38, chr17:70134929-71339950, duplication; **g**: Benko et al.[61], GRCh38, chr17:71072938-71767918, duplication). Genome segments (column) include SV segments (red borders) and distal genes within TAD (no borders). Colors of column names represent values of $p_{sv-gene}$ ranging from 0 (blue) to 1 (red). Source data are provided as a Source Data file.

three pathogenic SVs that are known to indirectly impact the downstream gene of *SOX9* with diverse phenotypes[59–61]. Here, PhenoSV correctly predicted SVs' functional impacts on *SOX9*, with $p_{sv-gene}$ being 0.63, 0.61, and 0.74, respectively. Results show that important features within the segments of target genes span all six feature categories (Fig. 4e–g). In contrast, evolutionary constraints within SV segments, such as conservation state annotations, are the major forces to yield pathogenic predictions for *SOX9*, especially the genome segment between *KCNJ2* and *SOX9* (Fig. 4f). This is in line with previous research[59–61] that disruptions of a highly conserved genome region between *KCNJ2* and *SOX9* play a pivotal role in various *SOX9*-related diseases. Taken together, PhenoSV can identify the critical pathogenic genes under either direct or indirect effects of SVs and can contribute to the interpretation of SVs in disease association studies.

### PhenoSV determines genes associated with distinct phenotypes from large SVs

Large SVs that affect similar genome regions can often lead to various phenotypes, impacting different sets of genes. This phenomenon poses a significant challenge for conducting association studies and identifying disease-related genes from such large SVs, as it requires a

massive sample size to achieve sufficient statistical power. Using a curated dataset of SVs affecting overlapping genome regions and potentially associated with inherited diseases and cancers, we investigated how PhenoSV can aid in disease association studies and determining genes responsible for distinct phenotypes within large SVs.

We first applied PhenoSV to 123 dosage-sensitive rare copy number variant (rCNV) segments that are associated with at least one disorder derived from genome-wide meta-analyses[47] to further validate if PhenoSV gene scores can be used as a good measure of pathogenicity in disease association studies. As indicated in the original study, deletion segments with large effect sizes of penetrance are enriched for genes under strong constraints compared to those with small effect sizes of penetrance when using minimum LOEUF (loss-of-function observed/ expected upper bound fraction) per segment as gene constraints measurement. This enrichment pattern was not observed for duplication segments. We similarly found that highly penetrant deletion segments have significantly lower min LOEUF scores than incompletely penetrant deletion segments (Wilcoxon rank-sum test, High OR vs. Low OR: p value = 0.012), but not for duplication segments (Wilcoxon rank-sum test, High OR vs. Low OR: p value = 0.065; left panel of Fig. 5a).

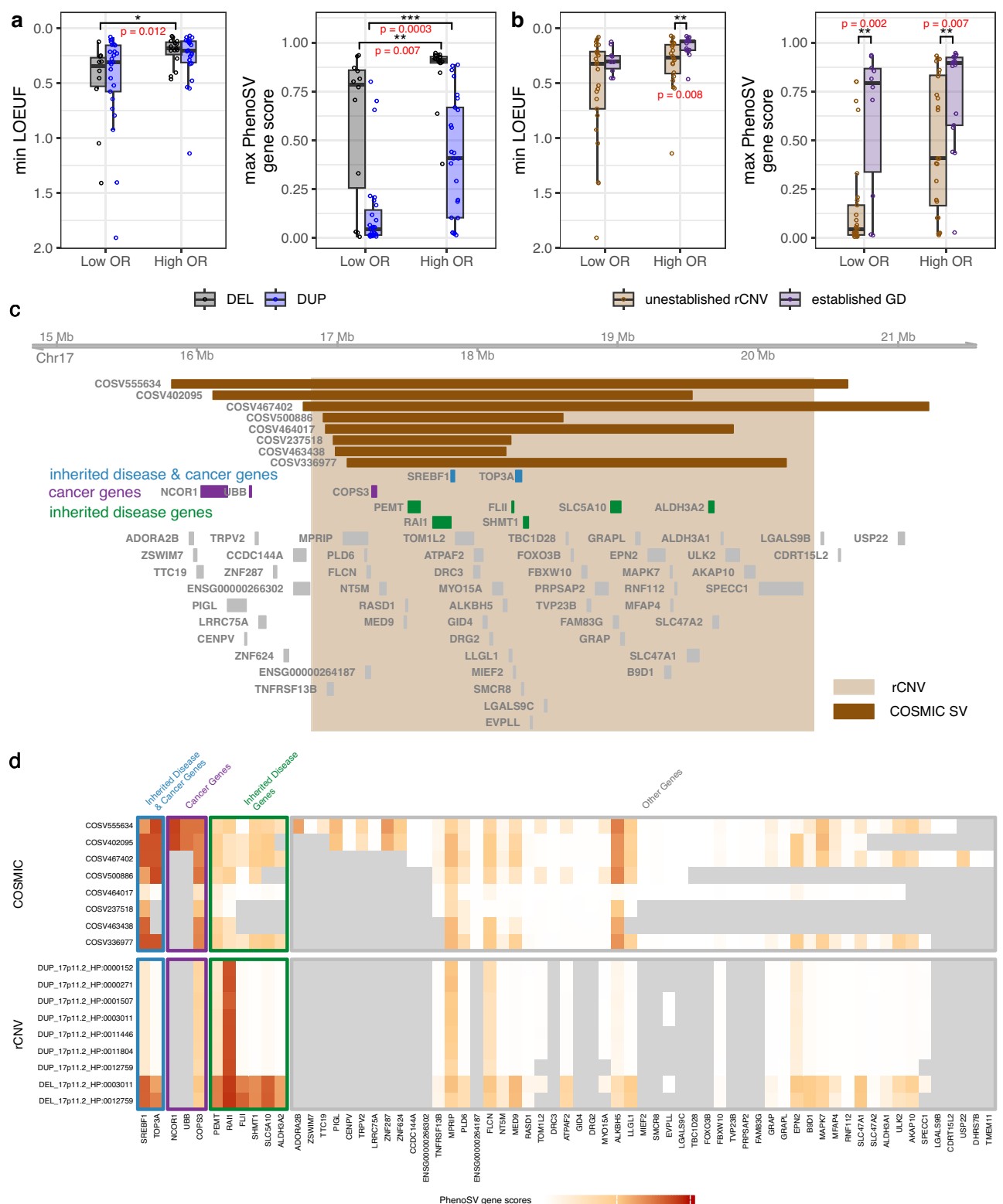

While when we measured gene constraints using maximum PhenoSV gene scores ($p_{sv-gene}$) of each segment, the enrichment pattern is more pronounced and significant for both deletions and duplications (Wilcoxon rank-sum test, High OR vs. Low OR, deletions: $p$ value = 0.007, duplications: $p$ value = 0.0003), and deletions segments have more pathogenic effects than duplication segments in both low-OR and high-OR groups (Wilcoxon rank-sum test, deletions vs duplications, Low OR: $p$ value = 0.0015, High OR: $p$ value < 0.0001; right panel of Fig. 5a). We further stratified these rCNVs by whether they are established GD

(genomic disorders) rCNVs, which are sites of recurrent rCNVs often formed by non-allelic homologous recombination and were compiled by the original study based on prior surveys. As shown in Fig. 5b, established GDs are significantly enriched for genes under stronger constraint in the high OR group (Wilcoxon test, $p$ value = 0.008) but not in the low OR group (Wilcoxon rank-sum test, $p$ value = 0.411) compared to unestablished rCNVs when using min LOEUF as the gene constraint measurement. Likewise, this trend becomes more pronounced and both significant for high-OR and low-OR groups when measuring gene

**Fig. 5 | PhenoSV predictions aid in identifying disease-related genes from SVs that are implicated in distinct phenotypes. a** Comparisons of genes covered by incompletely penetrant rCNV segments in the bottom third odds ratios (Low OR, $n = 12$ deletions and $n = 27$ duplications) and highly penetrant rCNV segments of the top third odds ratios (High OR, $n = 16$ deletions and $n = 25$ duplications). **b** Comparisons of genes covered by incompletely penetrant rCNV segments in the bottom third odds ratios (Low OR, 29 unestablished rCNV and 10 established GD) and highly penetrant rCNV segments of the top third odds ratios (High OR, $n = 24$ unestablished rCNV and $n = 17$ established GD). **a, b** Gene constraints were measured by minimum LOEUF (left panel) scores and maximum $p_{sv-gene}$ scores (right panel) of genes affected by each rCNV segment. Median (center line), IQR (box limits), and outliers (points) that exceeding 1.5x IQR were shown in the boxplot.

Two-sided Wilcoxon rank sum test $p$ value < 0.05*, <0.01**, <0.001***. **c** Displayed is an example of overlapping rCNV (chr17:16816686-20396687) and 8 COSMIC SVs affecting similar sets of genes, including genes associated with both inherited diseases and cancers, genes primarily associated with inherited diseases, genes mainly associated with cancers, and other genes. **d** Heatmap of gene-level PhenoSV predictions ($p_{sv-gene}^{phen}$) for 8 COSMIC SVs (upper panel) and 9 rCNV-phenotype pairs from (**c**). Genes associated with both inherited diseases and cancers, and genes mainly associated with cancers were given high scores for COSMIC SVs. Genes associated with both inherited diseases and cancers, and genes mainly associated with inherited diseases were given high scores for rCNVs. Source data are provided as a Source Data file. LOEUF loss-of-function observed/expected upper bound fraction, GD genomic disorder.

constraints by maximum PhenoSV gene scores (Wilcoxon rank-sum test, Low OR: $p$ value = 0.002, High OR: $p$ value = 0.007). The above results again demonstrate PhenoSV can prioritize pathogenic genes within large pathogenic SVs containing many genes.

We then applied PhenoSV and calculated $p_{sv-gene}^{phen}$ scores for 377 rCNV-phenotype pairs (consisting of 97 rCNV segments) and 2847 COSMIC CNVs based on HPO terms of the corresponding cancer types (Table S2). These rCNV segments and COSMIC CNVs have extensive overlap affecting similar sets of genes, yet leading to distinct phenotypes that are either inherited diseases or cancers. We got the top 100 genes associated with inherited diseases and the top 100 genes associated cancers, respectively, ranked by the maximum $p_{sv-gene}^{phen}$ scores of each gene affected by different SV-phenotype pairs. These top genes can be categorized into three types, that are: (1) 32 overlapping genes involved in both inherited diseases and cancers, such as *NF1* (relates to both Neurofibromatosis type 1[62] and Leukemia[63]), (2) 68 inherited diseases genes (e.g., *RAI1* and *NRXN1*), and (3) 68 cancer genes (e.g., *ATR* and *RAF1*). We then selected rCNVs and COSMIC CNVs that have overlap and simultaneously affect these three types of genes. We identified 4 rCNV segments (16p11.2, 17p11.2, 17q11.2, and 18p11.23-p11.32) and 123 corresponding COSMIC SVs (Table S3, 20 coding SVs and 103 noncoding SVs). Figure 5c displays the rCNV segment of 17p11.2 that is associated with several phenotypes of inherited diseases and overlaps with 8 COSMIC SVs, indicating that SVs in this region may lead to diverse phenotypes by different genes. The genes that are involved in both inherited diseases and cancers (e.g., *SREBF1* and *TOP3A*) were given to high $p_{sv-gene}^{phen}$ scores for both rCNVs and COSMIC CNVs. The genes primarily involved in cancers (e.g., *COPS3* and *NCOR1*) were given to high $p_{sv-gene}^{phen}$ scores for COSMIC CNVs but not for rCNVs. Likewise, the genes that are primarily involved in inherited diseases (e.g., *PEMT* and *RAL1*) were given to high $p_{sv-gene}^{phen}$ scores for rCNVs but low scores for COSMIC CNVs (Fig. 5d). Collectively, PhenoSV has unique advantage in detangling complex SV-phenotype associations and determining genes responsible for different phenotypes affected by SVs.

## Discussion

While long-read sequencing technologies significantly improved the number of detectable SVs per genome, it remains challenging to identify pathogenic SVs and assess the impacts of individual genes affected by the SVs. In this study, we developed PhenoSV, a phenotype-aware and interpretable model to score and prioritize disease-related SVs. Unlike most existing machine learning-based methods that only work for a subset of SV types, PhenoSV can interpret all major types of SVs, including deletions, duplications, insertions, inversions, and translocations, regardless of whether the SVs are coding or noncoding. The interpretability of PhenoSV enables the identification of critical genes affected by SVs and important genomic features linked to pathogenicity, making it an invaluable tool for disease research. Through extensive testing on labeled SV datasets, PhenoSV demonstrates improved predictive accuracy on the functional impacts of SVs than existing methods, especially for noncoding SVs. Applications in unlabeled disease datasets suggest the superior interpretability of

PhenoSV, which can determine disease-related genes affected by SVs, including genes within and outside SV-affected genomic regions responsible for different phenotypes.

We believe that this improved predictive performance and model interpretability result from three aspects of PhenoSV. First, we have overcome the data limitation by matching pathogenic noncoding SVs with their closest common noncoding SVs and training them together with a vast number of coding SVs. Due to the positional proximity, the regulatory information learned from coding SVs can significantly aid in predicting the functional impacts of noncoding SVs on distal genes. In comparison, existing approaches typically separate different types of SVs into different categories for model building, potentially losing the ability to borrow information across SV types. Second, we have integrated hundreds of genomics features across six functional categories for each genome segment impacted. This enables us to capture a wide range of information with a complete picture of the functional impacts of SVs on genes, across a diverse range of tissue types or cell types. Third, the PhenoSV workflow and model architecture are specially designed to address challenges in predicting the pathogenicity of SVs. To increase the interpretability of results on the gene level, PhenoSV dissects SVs into sequences of genome segments and models SV interpretation as a MIL problem. To make accurate genome-wide predictions, especially when inferring long-range functional impacts of noncoding SVs on distal genes, PhenoSV adopts a transformer-based architecture with two types of masked attention heads. These attention heads can learn two types of information separately: whether the given noncoding SV can alter expression levels of the target gene and whether the target gene is dosage sensitive to cause diseases. Taken together, these advancements render unique advantages to PhenoSV in identifying disease-related SVs and pinpointing critical genes that can be subject to manual examinations for further confirmation.

We also acknowledge several limitations of PhenoSV that can be improved in future research. In this study, we used aggregated features based on multi-experiment or multi-tissue samples for epigenetic activity[35] and cis-regulatory genome annotations[38,64]. Although SVs' impacts on genes may be tissue-dependent, much more labeled SVs with tissue-specific pathogenicity scores will be needed to train a model that can account for such information. We thus made the compromise of training a general model that is not tissue-specific, so that PhenoSV relies on pre-determined sets of candidate genes when interpreting the impacts of noncoding SVs, either by distance or TAD annotations. Since not all labeled SVs have the corresponding tissue source information and TAD annotations are tissue-dependent, we used a sub-optimal distance-based strategy to determine the candidate gene sets for all SVs during training. Nevertheless, tissue-specific TAD annotations can be used to derive more defined candidate gene sets when using PhenoSV. This capability is facilitated by the current command-line tool of PhenoSV, which enables users to employ their own TAD annotations, such as tissue-specific TAD, for specific analyses. As more experimental data, such as Hi-C data, becomes available and expand existing tissue-specific genome annotations, more efficient approaches will be explored for further improvements. In

addition, machine-learning-based models largely rely on existing labeled datasets for training and testing. Since the number of insertions, inversions, and translocations in the existing labeled datasets are too small to train a model directly, PhenoSV used deletions and duplications to approximate the impacts of these SV types to overcome data limitations. Labeled insertions, inversions, and translocations were only used as test datasets to evaluate the performance of PhenoSV and will not influence PhenoSV performance. However, we should acknowledge that the test dataset of inversions is still small, and users should be more cautious when interpreting results on inversions. Similarly, the number of noncoding SVs is limited in our training dataset. To train a model that can accurately predict the pathogenicity of noncoding SVs, we devised a strategy that makes the input features of coding SVs and noncoding SVs "look alike" (see Supplementary Materials). This approach enabled us to utilize information from coding SVs to enhance the training for noncoding SVs. Yet, the scarcity of pathogenic noncoding SVs in our training and testing dataset, along with the lack of tissue-specific functional annotations, remains a notable limitation that requires further improvement once appropriate datasets become available. Furthermore, to mitigate potential bias in the interpretation of SVs due to inherent distinctions between autosomes and sex chromosomes, as well as the absence of specific genomic features, such as gwRVIS and JARVIS, on sex chromosomes, our training efforts for PhenoSV only focused on autosomes. While our analysis results have shown that PhenoSV generalizes well in interpreting SVs on sex chromosomes, it is important to note that PhenoSV is trained on autosomes. Caution should still be exercised when interpreting SVs on sex chromosomes, including the sex of the individual and the mode of inheritance of the disease under consideration. Additionally, considering that the current labeled SV dataset primarily relies on the GRCh38 genome build, PhenoSV has naturally adopted the same genome reference. The advent of long-read sequencing technology, which enables the complete sequencing of the human genome[65], also accentuates the necessity of expanding PhenoSV's capabilities to encompass the T2T-CHM13 genome build and pangenome reference build, to interpret SVs in those newly sequenced regions. Finally, one additional practical limitation is that due to the use of a rich set of features in the machine-learning model, the file size for the feature set becomes very large (currently over 100 GB), making it difficult for typical users to take advantage of the model when they need to download a large file for local analysis. To address this concern, we have implemented a web server, so that users can use a web application to perform analysis on small set of SVs and examine results without command-line tools. As an additional alternative approach for users, we developed PhenoSV-light, which is a lightweight version of PhenoSV using only 42 carefully selected features with dramatically reduced file size (Table S5). Despite its reduced complexity, PhenoSV-light demonstrates comparable predictive accuracy to the original PhenoSV, except for translocations (see Supplementary Materials and Fig. S2). This practical alternative offers enhanced accessibility and usability of PhenoSV, and we have also implemented this functionality in the web server.

In summary, PhenoSV is a phenotype-aware machine-learning model that can accurately score and prioritize disease-related coding and noncoding SVs. PhenoSV has unique advantage in generating interpretable results on both the SV level and the gene level, which will significantly facilitate functional studies of SVs and fuel novel biological discoveries in disease research. For easy implementation, we provide a website server at https://phenosv.wglab.org and a command line tool at https://github.com/WGLab/PhenoSV[66].

## Methods

### Training, validation, and test datasets
We curated a large collection of human SV datasets from multiple sources to train and test PhenoSV, including ClinVar[42] (full release 02/ 2022), DECIPHER[43] (v11.15), SvAnna-curated dataset[26], a curated translocation dataset based on chimeric RNAs in non-diseased tissues[53] and COSMIC-curated fusion genes (release v97)[48], NCBI-curated common SVs (dbVar nstd186 03/08/2022 version), five LRS datasets[23,29,67–69], three disease cohorts (congenital abnormalities[44], autism[45], and epilepsy[46]), as well as a curated SV data with diverse phenotypes based on the germline rCNV (rare copy-number variants) data[47] and somatic COSMIC CNV data[48] (release v96). We define coding SVs as the ones overlapping with at least 1 bp on any exon of protein-coding genes according to the GENCODE v40 annotations[70], where only representative transcripts (tagged as basic) belonging to level1 (validated) or level2 (manual annotation) confidences were kept, otherwise the SVs are treated as noncoding SVs. It is important to note that coding SVs include SVs covering coding regions exclusively, as well as those covering both coding and noncoding regions. Conversely, noncoding SVs only cover noncoding regions. We use 0.01 as the threshold of minor allele frequency (MAF) to decide common SVs (MAF > 0.01). All SVs not in GRCh38 build were converted to GRCh38 assembly using CrossMap[71], and only SVs with over 99% mapping ratio were kept in our analysis.

We first collected SVs from ClinVar if they fulfill all the following requirements: (1) clinical significance of benign, benign/likely benign, likely pathogenic, pathogenic, pathogenic/likely pathogenic; (2) not somatic in origin; (3) SV type of copy number loss, deletion, copy number gain, duplication, insertion, or inversion; (4) SV size ranges from 50 bp to 1 Mbp; (5) best placement if there exists multiple placements per assembly. We labeled ClinVar SVs with clinical significance of pathogenic or likely pathogenic as pathogenic, and benign or likely benign as benign (Fig. S5). We similarly retrieved SVs from DECIPHER, SvAnna, NCBI curated common SVs, and LRS datasets mentioned above if they qualify: (1) SV size ranges from 50 bp to 1Mbp, and (2) SV type of deletion, duplication, insertion, or inversion. DECIPHER SVs with pathogenicity labels other than benign, likely benign, likely pathogenic, and pathogenic were excluded. We labeled pathogenic and likely pathogenic of DECIPHER SVs and all SvAnna SVs as pathogenic.

In this study, our primary focus is on autosomal SVs for training and testing PhenoSV. Unless explicitly stated otherwise, all datasets exclusively include autosomal SVs. For deletions and duplications, we compiled all coding SVs from ClinVar, DECIPHER, SvAnna, and five LRS datasets together, and all noncoding pathogenic SVs from ClinVar, DECIPHER, and SvAnna together. SV filtering and unification procedures were then performed to remove noises and redundancies. Coding SV deletions, coding SV duplications, noncoding SV deletions, and noncoding SV duplications were preprocessed separately. We set the reciprocal overlap threshold for SV filtering as 70% and SV unification as 90%, which are similar to previously used thresholds[26,31,72]. For SV filtering, we first excluded all common SVs from the compiled datasets if they overlapped with any NCBI-curated common SVs greater than the threshold of 70%. We then dropped both pathogenic and benign SVs with conflicting labels within three labeled datasets: ClinVar, DECIPHER, and SvAnna. We also removed rare benign SVs from unlabeled LRS datasets if they were identified as pathogenic in any of the three labeled datasets. For SV unification, we constructed an undirected SV network by connecting SV pairs if they had reciprocal overlap over 90%. Within each of the connected components of the network, we ordered SV primarily by starting position and secondarily by dataset sources of ClinVar, DECIPHER, SvAnna, and LRS datasets. We dropped the SVs with even indexes and repeated the steps from constructing the SV network to dropping SVs until there were no SV pairs exceeding reciprocal overlap threshold of 90%. After filtering and unification, there are 245 noncoding pathogenic SVs from ClinVar, DECIPHER, and SvAnna. We then matched each of these SVs with one NCBI-curated common SV, the closest noncoding SV with a similar set of candidate target genes. We combined this dataset of matched

noncoding SVs with all coding SVs from ClinVar that passed the filtering and unification procedure. We split these SVs, containing only deletions and duplications, into training, validation, and hold-out test datasets based on chromosomes prevent any information leakage and to ensure the reliability of our test results. The chromosome numbers for these splits were selected to produce balanced datasets concerning pathogenic SVs and benign SVs. The training set has 14,292 coding SVs (6609 pathogenic and 7683 benign) and 330 noncoding SVs (165 pathogenic and 165 benign) from chromosomes 1–10 and 17–22. The validation set has 2136 coding SVs (990 pathogenic and 1146 benign) and 46 noncoding SVs (23 pathogenic and 23 benign) from chromosomes 11–13. The hold-out test set has 2559 coding SVs (6609 pathogenic and 7683 benign) and 330 noncoding SVs (165 pathogenic and 165 benign) from chromosomes 1–10 and 17–22. Since we removed coding SVs that are almost identical to ClinVar coding SVs from DECIPHER and SvAnna curated dataset, potential information leakage can be avoided. All coding SVs from DECIPHER and SvAnna that passed the filtering and unification are used as independent test datasets: small (50 bp–100 kbp, 383 pathogenic and 366 benign) and large SV (100 kbp–1 Mbp, 1208 pathogenic and 801 benign) datasets (Supplementary Data 2). Furthermore, we compiled a sex chromosome test dataset comprising all SVs located on chrX and chrY, sourced from ClinVar, DECIPHER, and SvAnna. The SV filtering steps employed here are the same with those applied to autosomal chromosomes. In total, this dataset includes 2034 pathogenic SVs and 1934 benign SVs.

We compiled all SVs of insertions and inversions from ClinVar, SvAnna, and five LRS datasets together. To test PhenoSV performance for insertions and inversions, we randomly sampled the same number of insertions/inversions that can be treated as benign from LRS datasets to match pathogenic insertions/inversions from ClinVar and SvAnna. The test insertion dataset includes 175 pathogenic insertions (171 coding SVs and 4 noncoding SVs) and 175 benign insertions (171 coding SVs and 4 noncoding SVs). The test inversion dataset includes 20 pathogenic inversions and 20 benign inversions. We further curated a test translocation dataset, including 68 pathogenic fusion genes that appeared in more than 10 cancer samples from COSMIC database[48] and 38 common chimeric RNAs recurrent in all normal tissue types from a previous study[53]. The exact breakpoints for COSMIC translocations were estimated by the middle points of inferred intervals.

Three unlabeled SV datasets from human disease cohorts were curated as additional test datasets. Autosomal SVs of deletions and duplications larger than 50 bp were kept. For the congenital abnormality cohort, we used the CNV cohort data from a previous study[44]. We excluded patients with multiple potentially pathogenic SVs as annotated by the original study. For the autism cohort[45], we combined the discovery and the replicate datasets. For the epilepsy cohort, we only kept rare noncoding SVs. Finally, the congenital abnormality cohort contains 91 coding SVs, and the autism cohort has 222 noncoding SVs, with 126 being paternal and 96 being maternal. The epilepsy cohort has 373 SVs, with 150 SVs from epilepsy patients and 223 SVs from control individuals. Note that, most (>99%) of the noncoding SVs in the autism cohort and the epilepsy cohort have no overlap (defined previously as having 70% of bases that are covered by SVs in the training data) with noncoding SVs in the training dataset.

We also curated a dataset comprising germline rCNV segments[47] and somatic COSMIC CNVs[48] that affect overlapping genome regions but lead to distinct phenotypes. rCNVs potentially associated with inherited diseases are derived from genome-wide meta-analyses. COSMIC CNVs are compiled from tumor samples from patients with diverse types of cancer. We downloaded the consensus set of 178 disease-associated rCNVs from a previous study[47] and excluded the rCNVs exceeding the 70% threshold of reciprocal overlaps with any SVs in the training data. This leaves us 123 rCNV segments. We retrieved 89,057 autosomal SVs of deletions or duplications larger than 50 bp from COSMIC. In total, 83,503 SVs that can be mapped to specific HPO

terms based on their histology information were kept. We require rCNV segments to overlap with at least one COSMIC CNVs and similarly require COSMIC CNVs to overlap with at least one rCNV segment by setting the reciprocal overlap threshold as 70%. This final dataset contains 377 rCNV-phenotype pairs (97 rCNV segments) and 2847 COSMIC CNVs from 11 cancer types (Table S2).

## SV segmentation

We segment the genomic region of interest potentially affected by a given SV into a sequence of genomic segments. For a coding SV, we are interested in the genomic region covered by this SV. Thus, the sequence of genomic segments comprises all protein-coding genes and intergenic noncoding areas covered by the SV ordered by their starting positions. Note that it is possible to have two consecutive genic segments because some genes overlap, and each segment will represent the full-length gene regardless of the overlapping status. For a noncoding SV, the genomic region potentially affected by this SV includes the area within and outside this SV. In this study, we consider the 1 Mbp flanking regions of noncoding SVs unless explicitly mentioned otherwise. The segment sequence of a noncoding SV includes protein-coding genes within this pre-determined genomic region of interest, intergenic noncoding regions, and the noncoding SV itself, ordered by their starting positions. We add zero-padding segments as pseudo noncoding segments at the front and end of every genomic segment sequence to ensure there are noncoding segments for genes to attend to within the attention heads of type 1 (Fig. 1b). It's important to highlight that there exists a significant imbalance between the numbers of coding SVs and noncoding SVs in our training dataset. To mitigate the class imbalance issue, we designed the segmentation strategy above to balance between coding SVs and noncoding SVs in the input feature space. See Supplementary Materials for more detailed discussions.

## Feature annotation

We integrated 64 annotation types and performed one-hot encoding into 238 features that are used to annotate genomic segments. These 238 features can be grouped into six functional categories, including deleteriousness scores, epigenetic activities, disease constraints, genome element annotations, evolutionary constraints, and SV types (see Supplementary Data 1 for details). These features are functionally diverse with different genome coverage and resolution scales, including genome-wide, coding- and noncoding-region features, and at the locus, gene, or segment levels. For example, we incorporated segment-level genome annotation features, such as super-enhancers[73] and chromatin states[74]. We also included gene-level disease constraint features, such as dosage sensitivity-related scores of pHaplo and pTriplo[47]. This way, we could utilize these features collectively to account for the impacts on genes when disrupting genome regulatory elements. After downloading all 64 annotation types, we first used customized bash scripts to transform each feature into a bigwig track file using tools including BEDTools[75], BEDOPS[76], and UCSC bedGraphToBigWig[77]. All features in the GRCh37 build were converted into GRCh38 assembly using CrossMap[71]. We used genome copy number for the last feature dimension to represent different SV types. Deletions and duplications covered by SVs were encoding as 1 and 0, respectively. We encoded the SV type feature for genomic segments outside of SVs as 0.5, due to the lack of copy number changes (Fig. S1a). We performed one-hot encoding for all other categorical features, such as chromatin states, making the final annotation feature number being 238. For features having multiple scores on a single nucleotide position, such as CADD scores[78], we combined them into one score using the maximum one. Then, these bigwig track files were used to annotate each genome segment of each SV based on customized python scripts using assigned summary statistics for aggregation (Supplementary Data 1). For features having partial coverage within the genomic segment, we only aggregated feature scores on the region

that is covered. For instance, if the aggregation method is "mean", we divided the sum of the feature scores within the segment region by the covered feature length within the segment. We used zero score for features with no coverage within the segment region. Finally, all segment-wise feature scores were normalized using min-max scaling, that is $\frac{score - min(score)}{max(score) - min(score)}$, where the empirical min and max were estimated from SVs in the training dataset without being segmented.

## PhenoSV architecture and training

PhenoSV has two major components, a feature extractor and a classifier. The feature extractor takes a sequence of annotated genomic segments $\mathbf{X}_{SV} \in \mathbb{R}^{n_{sv} \times 238}$ and outputs an overall SV-level embedding and a list of gene-level embeddings. $n_{sv}$ is the number of genomic segments for a given SV, including two zero-padding segments. We first use two fully connected layers with dimensions of 800 and 512 to project $\mathbf{X}_{SV} \in \mathbb{R}^{n_{sv} \times 238}$ into $\mathbf{X}_{embed} \in \mathbb{R}^{n_{sv} \times 512}$, where each genomic segment is independent without information exchange. $\mathbf{X}_{embed}$ is then fed into four consecutive transformer blocks containing masked multi-head attention (MHA) layers with standard residual connection and layer normalization[40] to update sequence embeddings into those with contextual information, denoting as $\mathbf{X}_{out} \in \mathbb{R}^{n_{sv} \times 512}$. Relative positional encoding is used with maximum absolute distance being set as 20 segments[79]. We set the number of attention heads within each transformer block as 8. For each attention head $h$, the input feature matrix is linearly projected to $\mathbf{Q}^h$ (query), $\mathbf{K}^h$ (key), and $\mathbf{V}^h$ (value), each with a dimension of $n_{SV} \times 64$. Values represent the contextual information that each genomic segment injects into other segments. Queries represent the current information of a given genomic segment, and keys represent the information of the target genomic segments from which the query segment gets information. Then, the updated feature embedding for $i^{th}$ segment in head $h$ becomes: $\mathbf{Z}_{i,*}^h = \sum_{j=1}^{n_{SV}} a_{ij}(\mathbf{V}_j^h, * + \boldsymbol{\gamma}_{ij}^V)$, $Z^h \in \mathbb{R}^{n_{SV} \times 64}$. $\boldsymbol{\gamma}_{ij}^V$ is a vector of length 64, encoding pairwise relative positions between the $i$th and $j$th segment of distance $|i - j|$ for $\mathbf{V}^h$. Here, $a_{ij}^h = \frac{\exp e_{ij}^h}{\sum_{k=1}^{n_{SV}} \exp e_{ik}^h}$ is a normalized weight coefficient, controlling how much information the $i$th segment should get from the $j$th segment in the attention head $h$. The raw weight coefficient $e_{ij}^h$ is computed by $e_{ij}^h = \frac{\mathbf{Q}_{i,*}^h \cdot (\mathbf{K}_{j,*}^h + \boldsymbol{\gamma}_{ij}^K)^\top}{\sqrt{64}} + \mathbf{M}_{ij}^h$. Similar to $\boldsymbol{\gamma}_{ij}^V$, $\boldsymbol{\gamma}_{ij}^K$ is a relative positional embedding for $\mathbf{K}^h$. $\mathbf{M}^h \in \mathbb{R}^{n_{SV} \times n_{SV}}$ is a mask matrix that adjusts pairwise attention weights $e_{ij}^h$ between genomic segments based on the types of segment pairs. Intuitively, we would like elements of protein-coding genes to query information from either themselves to model direct effects or from target noncoding regions to model indirect effects. Thus, two types of mask matrices are designed to explicitly model two different attention mechanisms. For the first 4 heads modeling the indirect effects, $\mathbf{M}_{ij}^h = 0$ if (1) the $i$th segment is protein-coding genes, and the $j$th segment is either noncoding segments for coding SVs or SV-affected region for noncoding SVs or the zero-padding elements; or (2) the $i$th segment is noncoding regions, and $i = j$. Otherwise, $\mathbf{M}_{ij}^h = -999$. For the rest 4 heads modeling the direct effects, protein-coding genes only query information from themselves and $\mathbf{M}^h = I$, where $I$ is identity matrix. We drop all segments that are not protein-coding genes from $\mathbf{X}_{out}$ and use a max-pooling layer to aggregate gene-level embeddings into the overall SV embedding. The classifier contains a fully connected layer of dimension 128 and a logit classifier. It can take the SV-level embedding to generate PhenoSV score for SV-level pathogenicity, denoted as $p_{sv}$. The classifier can also take the list of gene-level embeddings and output a list of PhenoSV gene scores of $\{p_{sv-gene_1}, p_{sv-gene_2}, \ldots, p_{sv-gene_m}\}$, representing the pathogenicity of $m$ genes affected by the SV. Note that when a noncoding SV is intronic, the left and right adjacent segments

represent the same gene with two PhenoSV gene scores. We use the maximum one of these two scores for this gene's final PhenoSV gene score.

Although coding SVs and noncoding SVs were trained together, their confidence scores have intrinsically different scales, with noncoding SVs having high sensitivities but low specificities (Table S1). We conduct an additional calibration step within the logit space to provide pathogenicity scores with a probabilistic interpretation, allowing us to classify a call as pathogenic when the value of $p_{sv}$ or $p_{sv-gene}$ surpasses 0.5. Specifically, the raw confidence score of $p^0$ will be transformed as followed: $p^c = \text{sigmoid}(\log \frac{p^0}{1-p^0} - \log \frac{s}{1-s})$. $s$ is the optimal cutoff value with $s = 0.4934$ for coding SVs and $s = 0.7901$ for noncoding SVs, which are determined by Youden index in the hold-out test dataset. After calibration, we noticed a more balanced sensitivity and specificity for noncoding SVs (Table S1). When prior phenotype information is available, a list of phenotype-gene association scores will be generated using percentile-transformed Phen2Gene scores, whose minimum value is used in imputing gene scores outside the list. For an SV impacting $m$ genes, we denote phenotype-gene association scores as $\{p_{gene_1-phen}, p_{gene_2-phen}, \ldots, p_{gene_m-phen}\}$, and use the maximum value to define the phenotype-SV association score $p_{sv-phen}$. Then, the phenotype-aware PhenoSV score can be expressed by $p_{sv}^{phen} = p_{sv} \times p_{sv-phen}^\alpha$, and the phenotype-aware PhenoSV gene score is $p_{gene}^{phen} = p_{gene} \times p_{gene-phen}^\alpha$. Here, $\alpha$ is a parameter with non-negative values controlling for the degree of phenotype dependency. When $\alpha = 0$, $p_{sv}^{phen}$ is equal to $p_{sv}$ that is independent of phenotypes. When $\alpha$ is increased, $p_{sv}^{phen}$ becomes more reliant on phenotype associations.

Taken together, PhenoSV can predict functional consequences of a given SV by generating: (1) PhenoSV SV score $p_{sv}$ that represents the general pathogenicity of the SV, (2) PhenoSV gene score $p_{sv-gene}$ that represents the general pathogenicity of a specific gene affected by the SV, (3) phenotype-aware PhenoSV SV score $p_{sv}^{phen}$ that represents the pathogenicity of the SV relating to given phenotypes, and (4) phenotype-aware PhenoSV gene score $p_{sv-gene}^{phen}$ that represents the pathogenicity of a specific gene relating to given phenotypes affected by the SV (Fig. 1b).

PhenoSV was trained for 100 epochs with Adam optimizer by setting $\beta_1 = 0.9$, $\beta_2 = 0.98$ and $\epsilon = 10^{-9}$. The final model was chosen from the epoch when validation loss achieved the lowest during training. Cosine annealing with linear warmup was used for learning rate scheduler, with 20 epochs for warmup and the rest 80 epochs for cosine decay. The maximum learning rate is 0.0001. We used drop-out rate of 0.2 and the weight decay parameter of 0.0001 for model training regularizations. Batch size was set as 256, with accumulating gradients for every 4 batches.

## PhenoSV for insertions, inversions, and translocations

Although PhenoSV was trained using deletions and duplications, it can be adapted to SV types of insertions, inversions, and translocations. We treat deletions and duplications as the basic forms of SVs exerting functional impacts on genes and approximate the impacts of insertions, inversions, and translocations using deletions and duplications (Fig. 2), though the original feature profiles of these SV types can be different from those of proxies. For an insertion, we mainly consider its impacts on disrupting the local genomic element by treating it as a 100 bp deletion centered at the insertion breakpoint (Figs. 2b and S3). For an inversion, we consider the impacts of its two breakpoints and treat genes fully incorporated by the inversion region as unimpacted (Figs. 2c and S3). When the breakpoint of the inversion is located within a gene, regardless of an exon or an intron, we treat it as a deletion truncating the 3' side of the gene by the breakpoint because the promoters of affected genes and their 5' UTRs are intact. When the breakpoint of the inversion is located within a noncoding region, we treat it as a 100 bp deletion centered at the breakpoint as a proxy, similar to an insertion. Note that, when one breakpoint disrupts a gene

and the other disrupts noncoding region, we mainly consider the impacts of the first breakpoint that has direct impact on a gene. The overall pathogenicity for an inversion with two breakpoints is thus the maximum of the two $p_{sv}$. For a translocation, we always consider the impacts of both 5' and 3' breakpoints, which can potentially produce gene truncations and fusion genes, see Fig. 2d for more details.

## SV prioritization using PhenoSV

We performed SV prioritization using simulated SV profiles in patients, each with one true disease-related SV and ~19,000 background ones. We collected all pathogenic SVs that passed the abovementioned filtering and unification steps and with phenotype information, including 1007 coding SVs from ClinVar in the hold-out test dataset, 494 SVs from the independent test datasets of DECIPHER and SvAnna, all 193 noncoding SVs from ClinVar, DECIPHER, and SvAnna, and all 150 insertions and inversions from ClinVar and SvAnna. Each of those pathogenic SVs was treated as the true disease-related SV for one patient. Note that, for coding pathogenic SVs from DECIPHER, we only kept the ones that fully or partially contribute to a given patient's phenotypes. Due to the limited number of labeled pathogenic noncoding SVs, we incorporated all noncoding SVs we could gather. There may be a low level of information leakage present in noncoding SV prioritizations. We randomly sampled 25,000 background SVs by allele frequency from a background SV dataset for each SV profile. This background SV dataset has 364,961 SVs compiled from NCBI-curated common SVs and the long-read rare SVs, including deletions, duplications, inversions, and insertions. All allele frequencies of rare background SVs were imputed as 0.01. Since background SVs can overlap, we first clustered SVs and only randomly sampled one from each cluster for each final simulated SV profile. This is to ensure the simulated SV profiles are realistic without overlapping SVs. Thus, each patient's simulated SV profile contains ~19,000 background SVs with about 8000 being rare novel SVs. We first filtered out all common SVs (MAF < 0.01) from each of the simulated SV profiles and then used $p_{sv}^{phen}$ to prioritize SVs of each patient's profile by setting $\alpha = 0$, 0.2, 0.4, 0.6, 0.8, and 1, respectively. When setting $\alpha = 0$, $p_{sv}^{phen}$ is equivalent with $p_{sv}$ without using any phenotype information. Phenotype information in DECIPHER and SvAnna datasets is given in the format of HPO terms, whereas ClinVar provides MONDO, OMIM, ORPHA, MeSH, and HPO terms depending on SV sources. We mapped all phenotype terminologies into HPO terms that can be used as inputs of Phen2Gene. However, we clarify here that other phenotype-based gene prioritization tools[80–83] can also be used here as well, with simple reformatting of output files from such tools.

## Statistics and reproducibility

PhenoSV was trained on 2 NVIDIA A100 GPUs using Pytorch Lightning framework (1.6.4). We performed SV-wise annotations for both coding- and noncoding-SVs for the PhenoSV-XGBoost model. We conducted grid search using R package of caret (6.0.94)[84] to find the optimal set of hyperparameters using the same training and validation splits as the PhenoSV model, and XGBoost model was trained using R package of xgboost (1.7.5.1). The hyperparameters searched included the max depth of a tree (6, 10), percentage of features used (0.5, 0.75, 1), and the number of trees (100, 200). The optimal hyperparameters of the final model are max tree depth as 6, percentage of features as 0.75, and the number of trees as 100. Predictions of benchmark methods of CADD-SV, AnnotSV, and StrVCTVRE were obtained from the official web servers by uploading bed files containing test SV coordinates (CADD-SV: https://cadd-sv.bihealth.org; AnnotSV: https://lbgi.fr/AnnotSV/; StrVCTVRE: https://strvctvre.berkeley.edu). Predictions of SVScore (https://github.com/lganel/SVScore) were obtained by running the command line tool. Scores of SVSCORETOP10 were used as

SVScore predictions. Simulated SV profiles used for prioritizations were first transformed into VCF file format required by SvAnna (https://github.com/TheJacksonLaboratory/SvAnna) and then analyzed using the command line tool. All statistical analyses were performed in R, version 4.2.2. AUC values for model performances were calculated using R package of pROC (1.18.4)[85].

## Reporting summary

Further information on research design is available in the Nature Portfolio Reporting Summary linked to this article.

## Data availability

All relevant data supporting the key findings of this study are available within the article and its Supplementary Information files. The simulation and benchmarking data are provided at https://github.com/WGLab/PhenoSV[66]. The training and testing data can be accessed through ClinVar (full release 02/2022) [https://ftp.ncbi.nlm.nih.gov/pub/clinvar/vcf_GRCh38/archive_2.0/2022/], DECIPHER (v11.15) [https://www.decipergenomics.org], COSMIC (release v97 for gene fusion data and release v96 for somatic CNV data) [https://cancer.sanger.ac.uk/cosmic], dbVar (nstd186, nstd152, nstd162, nstd175) [https://www.ncbi.nlm.nih.gov/dbvar/studies] and Supplementary files of papers mentioned in the "Methods" section. Source data are provided with this paper.

## Code availability

The software with source codes is available at https://github.com/WGLab/PhenoSV[66]. A companion web server can be accessed at https://phenosv.wglab.org.

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

## Acknowledgements

We thank Mian Umair Ahsan for helpful comments and discussion and thank the IDDRC Biostatistics and Data Science core (HD105354) for consultation on machine-learning. This study is supported in part by NIH grant GM132713, HG013031 and HG013359 and the CHOP Research Institute. This study makes use of data generated by the DECIPHER Consortium (A full list of centers who contributed to the generation of the data is available from http://decipher.sanger.ac.uk/ and via email from decipher@sanger.ac.uk. Funding for the project was provided by the Wellcome Trust). COSMIC datasets used in this study is from Wellcome Sanger Institute that is operated by Genome research Limited and is available at https://cancer.sanger.ac.uk/cosmic.

## Author contributions

K.W. and Z.X. conceived the study. Z.X. collected the datasets, conducted data analysis, developed the method, software, and web server. K.W. supervised the study. Q.L. and K.W. tested the software and web server. Z.X., Q.L., L.M. and K.W. interpreted the results. Z.X. drafted the initial manuscript. All authors reviewed, edited, and approved the manuscript.

## Competing interests

The authors declare no competing interests.
