## [Peer Review File NEW · Nature Communications]

PhenoSV: Interpretable phenotype-aware model for the prioritization of genes affected by structural variantsReviewer #1 (Remarks to the Author):

The paper is in general very well written, and the authors proposed a phenotype-aware machine-learning model, PhenoSV, enabling the interpretation of all types of SVs and genes within and outside SV. And the proposed method outperforms existing methods and identifies pathogenic SVs responsible for different phenotypes, as well as critically important genes directly or indirectly affected by SVs, which allows the prioritization of candidate SVs from a large candidate list and facilitate the interpretation of disease association studies. Overall, this work is very meaningful. There are some questions, which should be considered before it is considered for publication.

1. In material and Methods, paragraph 2, the authors labeled pathogenic and likely pathogenic SVs as pathogenic, others as benign. As we know, the likely pathogenic SV may vary depending on the definition of disease group and the health group. Could we consider to either classify SV as three categories of pathogenic, likely pathogenic, and benign, or train a disease specific model.
2. I noticed the imbalanced class number distribution in training/validation/test dataset. In training set, there are significantly more coding SVs (14,292) compared to non-coding SVs (330). Moreover, there is an imbalance in SV types, with fewer insertions, inversions, and translocations due to smaller number of these types in human genome. In that case, PhenoSV under-performed in non-coding SVs compared to coding SVs (AUCs in Figure.3). Is there a way to enhance those imbalanced classes by computational approaches or simulated datasets?
3. In method part 'Training, validation, and test datasets', the authors filtered out common SVs from training set. However, some disease-risk SVs are germline SVs and commonly presented in large human cohorts. For example, the 32kb LCE3C/LCE3B deletion appears in 64% psoriasis-related samples and 55% control samples (Rafael de Cid et al. Nature Genetic, 2009). GnomAD (Nature, 2020) also reported that there are on average 133.4 pLoF SVs per genome while for rare SVs, there are only 5.5 pLoF SVs. So, if I understand correctly, there are many disease-related but common SVs, which are needed to be predicted or associated with disease risks. Is there any change to remove the filter process? Or when comparing performances, consider common SVs and rare SVs independently?
4. Another concern about those common SVs used for training process might overlapping with SVs in a new applied sample. Even rare SVs, token from multiple sources, are also possible to present in more than one sample. If one SV was chosen for training, and it also appears in a new-applied sample, the prediction result will be well but untrusted. It would be better to report any overlaps in training SVs with new-applied SVs (such as the 222 noncoding SVs in ASD dataset and 123 dosage-sensitive rCNV segments). This will make prediction results more convincing and reliable.
5. The web interface is very friendly for users since the large feature set and machine-learning model (PhenosvFile.tar, about 153GB). If possible, the authors could perform some model and feature interpretability analysis to transparentize how features and model involves in making decision. This will be helpful to reduce feature sizes and lightweight model architecture. Removal of non-decision-making features and redundant model parameters will not affect performances but result in much usability. Moreover, this will help to avoid the brute-force and black-box of utilizing machine-learning approaches.

Reviewer #2 (Remarks to the Author):

The authors present a nicely packaged machine learning method to simultaneously score the likely deleteriousness of structural variants and their constituent genes. The ability to derive gene-level scores is a very nice asset as more and more studies start accounting for the effect of structural variants. The multi-head attention architecture is a nice idea to model direct and indirect effect on the genes, and having a web browser and command line tools enable easy exploration and adoption of the prioritization information offered by this method. My queries and comments are below:

Major comments:

1. ChrX and Y are special - they were not included in this method. Though that is ok, some recognition of this fact should be made.
2. Why is the strategy for training, validation, and hold-out done by chromosome? Is there a

reason to avoid random sampling? By chromosome could introduce significant bias concerns due to genetic architecture differences by chromosome.

Minor comments:

1. Approximation of insertions, inversions, & translocations seem like a reasonable current implementation via proxy of deletions/duplications at break-ends. How sensitive is the approximation to the artificial 100bp size setting.
2. Why does the AUC of larger test SVs (100kb - 1mb) decrease compared to smaller test SVs, when larger SVs tend to be more pathogenic?
3. Why was allele frequency chosen not used for learning?
4. What source is the allele frequency annotation?
5. In portraying the value of PhenoSV in stratifying CRE-SVs in ASD, the conclusion is overstated. The test statistic should be comparing the transmission rate of predicted pathogenic CRE-SVs to either (a) predicated non-pathogenic or (b) overall transmission rate. If using (a), the effect size is very minimal such that there's no statistical significance.
6. Could we put a quantification on the extent of the significantly more epilepsy-related SVs as it pertains to the middle and right panel of Figure 4c?
7. How does the performance of this method compared to a normal fine-mapping approach for gene prioritization in the rCNV paper? Are there and how often are there segments/genes that this method can prioritize that does not have fine-mapping evidence?

Typos:

1. Formatting: Figure 4 sub panels should be renamed to reflect order in text

Reviewer #3 (Remarks to the Author):

In the current manuscript, the authors present a machine learning method (phenoSV) to identify pathogenic SVs by integrating diverse features. Briefly, phenoSV performs segmentation and annotation of SV regions and extracts 238 distinct features to train a transformer-based machine learning model. Authors claim that their approach outperforms some existing methods for SV prioritization. Although the study is of interest and timely, various major concerns need to be addressed, which are listed below.

1. The current classification of SVs as coding and non-coding could be clearer. Most larger SVs are likely to affect multiple coding and non-coding regions simultaneously. How does phenoSV handle these cases for prioritization?
2. Authors claim that phenoSV is unique compared to other methods as it can prioritize all categories of SVs (beyond deletions and duplications). To do this, they convert different categories of SVs into proxy deletion and duplication by applying certain approximations. How do these approximations influence the underlying feature profile of the original SV compared to their proxy? For instance, will the original insertion have a similar feature profile to the transformed proxy deletions?
3. SVs from certain chromosomes were used as hold-out and others as test sets. What was the rationale for this? Will the model performance remain intact upon randomly assigning SVs in these two groups?
4. Why is the non-coding SV training set small? How will it affect the performance of the model?
5. Why phenoSV performs worse for large coding SVs compared to smaller ones? Also, why genome-wide conservation score is not important for coding SVs?
6. Authors should report precision-recall curves for their model beyond auROC plots.
7. The number of SVs (beyond deletions and duplications) is very small, particularly for their non-coding SVs. How will this influence the performance of the phenoSV model?

8. Authors should comment on the potential issue of model overfitting with such a large amount of feature sets, many of which are likely to be highly correlated.

9. The cut-off for classifying pathogenic vs. benign SVs is very lenient, as reported for their transmission analysis. How will a higher threshold for pathogenic SVs influence the overall result?

10. The current phenoSV approach doesn't consider tissue-specificity for non-coding SVs. As discussed by the authors, this is one of the biggest drawbacks of phenoSV. Why can't authors utilize tissue-specific annotation or epigenomic signals in the current phenoSV framework?

Response to Comments

We thank reviewers for their generous and constructive comments on our manuscript titled “PhenoSV: Interpretable phenotype-aware model for the prioritization of genes affected by structural variants”. We have fully addressed these concerns, and we believe that the manuscript is substantially improved by addressing the reviewer’s comments. In addition to the additional analysis and results, we have also taken one reviewer’s suggestion seriously and established a light-version of the PhenoSV tool which results in ~10X reduction of file sizes to improve usability. The original reviewers’ comments are in **bold font** and our point-by-point responses are given below. The changes that we made in the main manuscript were highlighted in **red fonts**.

Reviewer #1

1. In material and Methods, paragraph 2, the authors labeled pathogenic and likely pathogenic SVs as pathogenic, others as benign. As we know, the likely pathogenic SV may vary depending on the definition of disease group and the health group. Could we consider to either classify SV as three categories of pathogenic, likely pathogenic, and benign, or train a disease specific model.

We thank the reviewer for the comments. The pathogenicity of SVs can indeed vary depending on disease definitions, and the labels in ClinVar dataset sometimes have ambiguities as well (There are labels of "benign," "benign/likely benign," "likely pathogenic," "pathogenic," "pathogenic/likely pathogenic," "uncertain significance," in addition to "conflicting interpretations of pathogenicity.") In response to the reviewer's suggestion, we have carefully considered the issue of treating SV classification as a multi-category task, as well as the training of disease specific models. **First**, treating SV pathogenicity labels as a binary variable is a commonly employed strategy in existing machine learning-based models, such as in CADD-SV, SVFX, and StrVCTVRE. The main reason is that binary classification can facilitate minimization of loss function and increase interpretability of models on a quantitative scale: Therefore, we focused our model training on distinguishing between pathogenic SVs and benign SVs using binary labels. However, this approach allows for a straightforward interpretation and the continuous confidence scores (ranging from 0 to 1) can be used to infer the probabilities of an SV being pathogenic in general, irrespective of disease types. In the next step, when phenotype or disease information is available, we use the information to further fine-tune the score to reflect the pathogenicity of variants with respect to specific diseases. To further address the reviewer’s question, we have now tested the quantitative score over the three categories “benign”, “likely pathogenic” and “pathogenic”, and we observed increasing trend of the predictive scores (**Figure S5**). This additional analysis is now described in the supplementary materials (section ‘**Ambiguities of pathogenicity labels with different disease definitions**’, page 2, paragraph 3). **Second**, training a disease-specific model with combined inputs of SV features and patient’s phenotype terms can be appealing, compared to our current procedure of training a general model and then fine-tune model output using phenotype terms (when available). We indeed experimented this strategy previously but encountered the practical problem that most of the training samples do not have corresponding phenotype information

(or even disease information). With reduced sample size, the results are not satisfactory even in cross validation settings. Furthermore, as there are almost 18,000 possible HPO terms, adding raw HPO terms greatly increased the model complexities even when phenotype embedding is used in the predictive model. Thus, we opt to a generic model and utilize extra genotype-phenotype associations (e.g. Phen2Gene) to infer SV-disease associations. This procedure has the advantage of working on both relatively common diseases with general disease descriptors (such as disease name only without HPO terms) and rare diseases with more specific phenotype terms (such as a list of HPO terms).

2. I noticed the imbalanced class number distribution in training/validation/test dataset. In training set, there are significantly more coding SVs (14,292) compared to non-coding SVs (330). Moreover, there is an imbalance in SV types, with fewer insertions, inversions, and translocations due to smaller number of these types in human genome. In that case, PhenoSV under-performed in non-coding SVs compared to coding SVs (AUCs in Figure.3). Is there a way to enhance those imbalanced classes by computational approaches or simulated datasets?

We thank the reviewer for raising this important question. This is due to the much better understanding and disease annotation of coding variants (which may directly disrupt gene products) versus noncoding variants (which may target regulatory regions that influence levels of gene expression). While we recognize the class imbalance of coding SVs vs noncoding SVs, we did not adopt simulation approaches to artificially generate structural variants, or under/over sampling strategies to balance the numbers of coding SVs and noncoding SVs. The main reason is that the performance of computational approaches critically depends on how the simulation is performed for noncoding variants, yet it is difficult to justify what is the appropriate simulation strategy for the pathogenicity of noncoding SVs. To address this problem, we made the coding SVs and noncoding SVs “look alike” in the input feature space to alleviate the class imbalance issue. Specifically, we segmented coding SVs into sequences of noncoding and coding regions that the SVs impact directly. If we only input the noncoding regions that noncoding SVs impact directly, the coding SVs and noncoding SVs are straightforward to be distinguished by the model through features such as exon annotations. Thus, for noncoding SVs, we segmented coding and noncoding regions within a given distance or TAD (see Methods, SVs segmentation). Only masks of attention heads between coding SVs and noncoding SVs are different. In this way, we essentially incorporated into the model the information that noncoding SVs learned from a large number of coding SVs. To further address the reviewer’s comments and explain this strategy, we have expanded the description in Methods (session ‘**SV segmentation**’, page 18, paragraph 2) and also added additional discussions in the supplementary material (section ‘**Imbalanced number of coding SVs and noncoding SVs for training**’, page 3, paragraph 2).

In Methods, SV segmentation session:

We add zero-padding segments as pseudo noncoding segments at the front and end of every genome segment sequence to ensure there are noncoding segments for genes to attend to within the attention heads of type 1 (Figure 1b). *It’s important to highlight that there exists a significant imbalance between the numbers of coding SVs and noncoding SVs in our training dataset. To*

mitigate the class imbalance issue, we designed the segmentation strategy above to balance between coding SVs and noncoding SVs in the input feature space. See supplementary materials for more detailed discussions.

3. In method part ‘Training, validation, and test datasets’, the authors filtered out common SVs from training set. However, some disease-risk SVs are germline SVs and commonly presented in large human cohorts. For example, the 32kb LCE3C/LCE3B deletion appears in 64% psoriasis-related samples and 55% control samples (Rafael de Cid et al. Nature Genetic, 2009). GnomAD (Nature, 2020) also reported that there are on average 133.4 pLoF SVs per genome while for rare SVs, there are only 5.5 pLoF SVs. So, if I understand correctly, there are many disease-related but common SVs, which are needed to be predicted or associated with disease risks. Is there any change to remove the filter process? Or when comparing performances, consider common SVs and rare SVs independently?

Because common SVs are less likely to be pathogenic than rare SVs, filtering out common SVs can help decrease false positive rate in the training dataset. Moreover, by removing common SVs, PhenoSV is steered to capture features that distinguish SV pathogenicity, rather than being confounded by the distinction between rare and common SVs. The challenge in clinical interpretation of SVs is to identify highly penetrant variants with large effect sizes, rather than finding disease associated polymorphisms as they usually serve as proxy markers for another disease-casual genetic mutation within a linkage disequilibrium block.

We thank the reviewer for bringing up the issue on LCE3C/LCE3B deletions. We also investigated the 32kb LCE3C/LCE3B deletion (chr1:152583066-152615265) as mentioned. This deletion was not in our training dataset, and PhenoSV predicted this SV to be benign with a score of 0.009 (when examining genes separately: LCE3C has a score of 0.008 while LCE3B has a score of 0.011). We then searched the genome region of this deletion in ClinVar and found a 203kb copy number loss that covers the entire 32kb region and contains both LCE3C/LCE3B, yet this copy number loss is asserted as being benign (VCV000152664.1, chr1: 152526704-152729716, GRCh38). Therefore, this SV (which is a genetic polymorphism) may be associated with diseases with small effect sizes of OR=1.4, but a complete loss of the region does not impact disease status. To further address the reviewer’s comments, we have added discussions in supplementary materials (session ‘**Interpretations of disease-associated common SVs**’, page 4, paragraph 2), indicating the focus on highly penetrant structural variants that may directly impact genome function.

4. Another concern about those common SVs used for training process might overlapping with SVs in a new applied sample. Even rare SVs, token from multiple sources, are also possible to present in more than one sample. If one SV was chosen for training, and it also appears in a new-applied sample, the prediction result will be well but untrusted. It would be better to report any overlaps in training SVs with new-applied SVs (such as the 222 noncoding

SVs in ASD dataset and 123 dosage-sensitive rCNV segments). This will make prediction results more convincing and reliable.

Thank you for the comment and this is indeed an important factor to consider in the analysis. For the rCNV segment dataset, we excluded those SVs with over 70% reciprocal overlap with any SVs in our training dataset (see **Materials and Methods**, page 17 paragraph 4). For the epilepsy and autism datasets, we tried to use the same strategy to filter out noncoding SVs with over 70% reciprocal overlap with any noncoding SVs of the training dataset. Because the number of noncoding SVs we used for training is small, we did not find any noncoding SVs in these two datasets exceeding the threshold. To further address this concern, we now define percentage overlap as the percentage of bases of an SV that can be covered by any SVs in pre-defined dataset. The average numbers of percentage overlap in epilepsy and autism datasets with training noncoding SVs are 0.54% and 0.25%, respectively. In comparison, the average percentage overlap of rCNV segments with all training SVs is 29%. Therefore, the statistics show that the test results in these unlabeled datasets are reliable. We clarified this point in the revised manuscript (in **Materials and Methods**, session '**Training, validation, and test datasets**', page 17 paragraph 3).

In Materials and Methods, Training, validation, and test datasets:

The epilepsy cohort has 373 SVs, with 150 SVs from epilepsy patients and 223 SVs from control individuals. Note that, most (>99%) of the noncoding SVs in the autism cohort and the epilepsy cohort have no overlap (defined by having 70% of bases that are covered by SVs in training data) with noncoding SVs in the training dataset.

5. The web interface is very friendly for users since the large feature set and machine-learning model (PhenosvFile.tar, about 153GB). If possible, the authors could perform some model and feature interpretability analysis to transparentize how features and model involves in making decision. This will be helpful to reduce feature sizes and lightweight model architecture. Removal of non-decision-making features and redundant model parameters will not affect performances but result in much usability. Moreover, this will help to avoid the brute-force and black-box of utilizing machine-learning approaches.

We thank the reviewer for this valuable comment, which prompted us to make substantial changes to increase the usability of the model. Detailed descriptions of the additional analysis are now documented in the revised manuscript (**Discussion**, page15, paragraph1; **supplementary materials**, session '**PhenoSV-light training and testing**', page1, paragraph 1). Specifically, we first performed performance evaluations and recognized that the most time-consuming part for running PhenoSV is the annotation step, which involves querying 238 features for all genome segments impacted by SVs (this step is bound by disk I/O operations using the genome wig files). To address this hurdle, we carefully analyzed our feature importance results of PhenoSV (Figure 3e-g) and used this information as a guide to select a subset of features to train a lightweight version of PhenoSV, aptly named PhenoSV-light. PhenoSV-light is trained with only 42 important features, making the annotation step more

efficient. Specific considerations when selecting the PhenoSV-light feature set lie in two folds: (1) We ensured that the chosen features encompass all five categories, as it has been established that all feature categories together yield the best results (Figure 3g). (2) Specifically, we selected the top 5 important features from each category for both coding SVs and noncoding SVs, leading to the derivation of the 42 important feature set (Figure 3e-f). We then compared model performance between PhenoSV and PhenoSV-light using the same test datasets in the main manuscript. Based on our results, PhenoSV-light demonstrates largely comparable prediction accuracy to PhenoSV, except for translocations (Figure S2). This indicates that PhenoSV-light offers a highly efficient alternative for most SV types with minimal compromise in predictive accuracy. Accordingly, we revised the discussion part of our manuscript, with more details about PhenoSV-light in supplementary materials. Finally, we have made modifications to the web server to use the light version optionally.

In Discussion:

Finally, one additional “practical” limitation is that due to the use of a rich set of features in the machine-learning model, the file size for the feature set becomes very large (currently over 100GB), making it difficult for typical users to take advantage of the improved model due to the need to download a large file for occasional local analysis. To address this concern, we have made a web interface for these groups of users, so that they can use a web application to perform analysis on small set of SVs and examine results without command-line tools. As an additional alternative approach for users, we developed PhenoSV-light, which is a lightweight version of PhenoSV using only 42 carefully selected features with dramatically reduced file size (Table S7). Despite its reduced complexity, PhenoSV-light demonstrates comparable predictive accuracy to the original PhenoSV, except for translocations (see supplementary materials and Figure S2). This practical alternative offers enhanced accessibility and usability of PhenoSV, and we have also implemented this functionality in the web server.

Reviewer #2

Major comments:

1. ChrX and Y are special - they were not included in this method. Though that is ok, some recognition of this fact should be made.

Thank you for raising this comment. We acknowledge the intrinsic differences between autosomes and sex chromosomes in the interpretation of structural variants, which may introduce biases if such differences are not appropriately addressed. Moreover, some features such as JARVIS and gwRVIS lack scores for sex chromosomes. In light of these considerations, in the previous version of the manuscript, we did not include chrX/chrY to ensure the reliability and validity of our findings. Instead, we focused our training efforts for PhenoSV on autosomes. Nevertheless, we completely agree with the reviewer that it is important to address sex chromosomes in genomic analyses. Therefore, we have now performed exploratory analysis to use the autosomal models to interpret chrX/chrY variants (we had to treat some of the missing features with imputed values). We found that PhenoSV that is trained on autosomes performed

well in sex chromosomes with AUC of 0.94 (95% CI: 0.93 – 0.95). In the revised manuscript, we have described this additional analysis in revised manuscript (**Results**, session '**PhenoSV accurately predicts pathogenicity of both coding SVs and noncoding SVs**', page 7 paragraph 1), and added discussions on sex chromosomes (**Discussion**, page 14 paragraph 2).

In Results, PhenoSV accurately predicts pathogenicity of both coding SVs and noncoding SVs:

More importantly, PhenoSV can generate interpretable results by making predictions on the gene level, which is a common limitation for traditional machine-learning methods. Notably, we observed lower AUCs of large SVs than small SVs in the independent test set for all models except CADD-SV (Figure 3b-c). This performance decrease for larger SVs could be attributed to potential ascertainment biases arise from disparities in the techniques used for SV detection (see supplementary materials). Additionally, we evaluated PhenoSV's performance in interpreting SVs located on sex chromosomes (2,034 pathogenic and 1,934 benign SVs). As shown in Figure S6, PhenoSV generalizes well for SVs on sex chromosomes, achieving an AUC of 0.94 (95% CI: 0.93-0.95).

2. Why is the strategy for training, validation, and hold-out done by chromosome? Is there a reason to avoid random sampling? By chromosome could introduce significant bias concerns due to genetic architecture differences by chromosome.

Since structural variants can overlap with each other or influence each other (for example, when residing in the same TAD) in the same chromosome, splitting the dataset into training/validation/hold-out data sets by chromosomes is a commonly used strategy in literature to prevent information leakage from training dataset to validation/hold-out test set. For example, StrVCTVRE used leave-one-chromosome-out for cross validation, and also used chr1, chr3, chr5, and chr7 from ClinVar as the hold-out test set. We certainly acknowledge the existence of genetic architecture differences by chromosomes; here the hypothesis is that if the model trained on specific chromosomes can perform well in other chromosomes, then the model must have extracted useful features that are not dependent on chromosomes. To further assess the validity of the strategy of splitting by chromosomes, as suggested by the reviewer, we also conducted experiments using random splitting of the dataset. As shown in Table S8, random splitting led to improved performance in the hold-out test dataset for both coding SVs (AUC of random split: 0.948; AUC by chromosome: 0.911) and noncoding SVs (AUC of random split: 0.89; AUC by chromosome: 0.86), On the other hand, the performance in the independent test datasets for small SVs (AUC of random split: 0.876; AUC by chromosome: 0.874) and large SVs (AUC of random split: 0.769; AUC by chromosome: 0.770) remained nearly identical for both splitting strategies. Despite these results, it is important to acknowledge that random splitting may lead to inflated performance results within the hold-out test set due to some extent of information leakage. To clarify the rationale of splitting by chromosomes, we revised the methods part of the manuscript (**Materials and Methods**, session '**Training, validation, and test datasets**', page 16 paragraph 2), and we also added results of using the

random split strategy in the supplementary materials (**supplementary materials**, section '**Splitting datasets by chromosomes and splitting datasets by random**', page 1, paragraph 3).

In Materials and Methods, Training, validation, and test datasets:

We split these SVs, containing only deletions and duplications, into training, validation, and hold-out test datasets based on chromosomes ~~that can produce almost balanced datasets~~ to prevent any information leakage and to ensure the reliability of our test results. The chromosome numbers for these splits were selected to produce balanced datasets concerning pathogenic SVs and benign SVs. The training set has 14,292 coding SVs (6,609 pathogenic and 7,683 benign) and 330 noncoding SVs (165 pathogenic and 165 benign) from chromosomes 1-10 and 17-22.

Minor comments:

1. Approximation of insertions, inversions, & translocations seem like a reasonable current implementation via proxy of deletions/duplications at break-ends. How sensitive is the approximation to the artificial 100bp size setting.

We thank the reviewer for this question. To address the question, we have performed sensitivity analysis to study how window sizes influence PhenoSV predictions. We focused on the test dataset of insertions to compare PhenoSV predictions using various window sizes: 50bp, 100bp, 150bp, 200bp, 300bp, and 500bp, since translocations and coding inversions do not use proxies with specific window sizes. Our findings revealed that PhenoSV predictions were highly correlated across different window size settings (**Figure S3**). The results demonstrate the robustness and stability of PhenoSV predictions when choosing different window sizes for SV proxies. We added the sensitivity analysis into supplementary materials (supplementary materials, session '**Sensitivity analysis of window size selection for SV proxies**', page1, paragraph 2).

2. Why does the AUC of larger test SVs (100kb - 1mb) decrease compared to smaller test SVs, when larger SVs tend to be more pathogenic?

We initially had the same question as the reviewer. As shown in Figure 3b-c, all models except for CADD-SV yielded lower AUCs in the test set of large SVs. This finding aligns with the results presented in the StrVCTVRE paper, where the authors reported higher AUCs for SVs categorized as either small (<30kb) or large (>500kb) than those with mid-length (30kb-500kb). The model performance decrease for larger SVs could be attributed to potential ascertainment biases between the test dataset of small SVs and the test dataset of large SVs, due to the differences in SV detection technologies. Specifically, larger SVs with sizes over 100kbp are primarily detected by microarrays (with imprecise breakpoints) and are more likely to be reported in literature. On the other hand, smaller SVs, ranging from 50bp to 100kbp, are commonly identified using Next-Generation Sequencing (NGS) techniques; most of these SVs are not reported in literature or documented in databases unless there is clear evidence for pathogenicity. Since previous studies did not specifically discuss this issue, we have now added discussions in the revised manuscript (**Results**, session '**PhenoSV accurately predicts**

pathogenicity of both coding SVs and noncoding SVs', page 7 paragraph 1) and supplementary materials (session 'Potential ascertainment biases between the test dataset of small and large SVs', page 5, paragraph 2).

In Results, PhenoSV accurately predicts pathogenicity of both coding SVs and noncoding SVs: More importantly, PhenoSV can generate interpretable results by making predictions on the gene level, which is a common limitation for traditional machine-learning methods. Notably, we observed lower AUCs of large SVs than small SVs in the independent test set for all models except CADD-SV (Figure 3b-c). This performance decrease for larger SVs could be attributed to potential ascertainment biases arise from disparities in the techniques used for SV detection (see supplementary materials). Additionally,...

3. Why was allele frequency chosen not used for learning?

It is well known that allele frequency inversely correlates with the functional significance of mutations, due to purifying selection pressure. We intentionally omitted allele frequency as a feature in training PhenoSV to avoid potential ascertainment biases when selecting training dataset. Instead, since allele frequency information was not part of the features used in PhenoSV's training, we can employ allele frequency as a performance metric to assess whether PhenoSV scores exhibited expected negative correlation with allele frequency. This analysis provided insights into PhenoSV's predictive capabilities and its association with allele frequency in the context of pathogenicity prediction. In response to this comment, we have now added discussions in the revised manuscript (**Results**, session 'PhenoSV accurately predicts pathogenicity of both coding SVs and noncoding SVs', page 7 paragraph 2).

In Results: PhenoSV accurately predicts pathogenicity of both coding SVs and noncoding SVs
Due to the presence of purifying selection pressure, allele frequency is expected to inversely correlate with the functional significance of mutations⁵¹. To avoid potential ascertainment biases during the selection of training dataset, we deliberately excluded allele frequency from the input feature set during the training of PhenoSV. However, we can now employ allele frequency as a performance metric to evaluate whether predicted PhenoSV scores exhibited the anticipated negative correlation with allele frequency in the hold-out test set. The estimation of allele frequencies was carried out based on gnomAD-SV database⁵². As expected, PhenoSV scores are negatively correlated with SV allele frequency, where rarer SVs are more likely to be pathogenic (Spearman's rho = -0.19, p-value <0.0001).

4. What source is the allele frequency annotation?

Thanks for pointing out. We used gnomAD-SV for estimating SV allele frequency. To clarify this, we revised the manuscript (**Results**, session 'PhenoSV accurately predicts pathogenicity of both coding SVs and noncoding SVs', page 7 paragraph 2).

In Results: PhenoSV accurately predicts pathogenicity of both coding SVs and noncoding SVs

Due to the presence of purifying selection pressure, allele frequency is expected to inversely correlate with the functional significance of mutations⁵¹. To avoid potential ascertainment biases during the selection of training dataset, we deliberately excluded allele frequency from the input feature set during the training of PhenoSV. However, we can now employ allele frequency as a performance metric to evaluate whether predicted PhenoSV scores exhibited the anticipated negative correlation with allele frequency in the hold-out test set. The estimation of allele frequencies was carried out based on gnomAD-SV database⁵². As expected, PhenoSV scores are negatively correlated with SV allele frequency, where rarer SVs are more likely to be pathogenic (Spearman's rho = -0.19, p-value <0.0001).

5. In portraying the value of PhenoSV in stratifying CRE-SVs in ASD, the conclusion is overstated. The test statistic should be comparing the transmission rate of predicted pathogenic CRE-SVs to either (a) predicated non-pathogenic or (b) overall transmission rate. If using (a), the effect size is very minimal such that there's no statistical significance.

We agree with the reviewer. We compared the transmission rate between predicted pathogenic SVs with predicted benign SVs. Due to the limited sample size of the cohort, the effect size difference (71% vs 64%) between pathogenic paternal SVs and benign paternal SVs is not statistically significant. To address this comment, we toned down our conclusion and revised the result section (**Results**, session '**PhenoSV determines disease-related genes indirectly affected by noncoding SVs**', page 11 paragraph 1).

In Results: PhenoSV determines disease-related genes indirectly affected by noncoding SVs

After stratifying CRE-SVs into pathogenic ($p_{sv-gene} \geq 0.5$) and benign ($p_{sv-gene} < 0.5$) groups using PhenoSV, we observed the over-transmission pattern of paternally inherited CRE-SVs more evident for predicted pathogenic ones (29/41; transmission rate = 71%; binomial test p-value = 0.01) than predicted benign ones (38/59; transmission rate = 64%; binomial test p-value = 0.04); a slightly larger effect size of over-transmission pattern was observed for paternally inherited pathogenic SVs (29/41; transmission rate = 71%; binomial test p-value = 0.01) than benign SVs (38/59; transmission rate = 64%; binomial test p-value = 0.04). Although statistical significance was not achieved due to limited sample sizes (pathogenic SVs vs benign SVs, proportion test p-value=0.656), these results suggest the values of PhenoSV in determining pathogenic genes indirectly affected by noncoding SVs. When classifying pathogenic CRE-SVs and benign CRE-SVs, different thresholds of $p_{sv-gene}$ (such as top and bottom 30% quantiles) can be used and we found that different thresholds do not influence the overall conclusion of the analysis (supplementary materials, **Table S9-S10**).

6. Could we put a quantification on the extent of the significantly more epilepsy-related SVs as it pertains to the middle and right panel of Figure 4c?

As observed in the middle and the right panel of Figure 4c, epilepsy-related SVs are enriched in the patient group compared to the control group. To quantify this enrichment, we used the enrichment score, which is defined as integrated cumulative affected sample numbers of patients over controls (represented by areas of orange shades in Figure 4d). For this calculation, we used the upper bound of the controls' 95% confidence intervals. The enrichment scores for overall SV, max epilepsy gene, and the closest epilepsy gene are 9.05, 51.24, and 18.36, respectively.

The statistics confirms our conclusion in the main manuscript that there are significantly more epilepsy-related SVs in the patient group, which do not necessarily affect the nearest epilepsy genes. Accordingly, we revised the Figure 4d (original Figure 4c) by annotating the enrichment score.

In Figure 4d:

(d) Displayed are cumulative affected sample numbers (y-axis) with p_{sv} (left panel), $p_{sv-gene}$ of the most affected epilepsy gene (middle panel), and $p_{sv-gene}$ of the nearest epilepsy gene (right panel) larger than given thresholds (x-axis) of 150 patients (red line) and 150 controls on average (blue line). Confidence intervals of controls (blue shades) are calculated by randomly sampling 150 samples from 223 controls for 100 times. Area of orange shades represent enrichment score, defined as integrated cumulative number of affected patient samples over upper bound of the 95% confidence interval of controls.

7. How does the performance of this method compared to a normal fine-mapping approach for gene prioritization in the rCNV paper? Are there and how often are there segments/genes that this method can prioritize that does not have fine-mapping evidence?

The rCNV paper derived gene-level pHaplo and pTriplo scores; these are gene-level scores that are not related to specific mutations or specific phenotypes. On the other hand, gene-level PhenoSV scores would vary among different structural variants impacting the genes and patients' phenotypes. Thus, these two methods are not directly comparable.

To address the second question, we take the SV case we investigated in the main manuscript as an example (Figure 4b, chr16: 28473235-30186830, deletion, GRCh38). This SV deletes the entire KIF22 gene. The original paper (Middelkamp, S. et al. Genome Med, 2019) predicted this gene to be the tier3-level driver gene for the SV pathogenicity. PhenoSV predicted the

pathogenicity score as 0.92 in general and 0.72 with phenotype information. While the pHaplo score of this gene is only 0.15, indicating deletion tolerance (pHaplo scores ≥ 0.86 indicate that the average effect sizes of deletions are as strong as the loss-of-function of genes known to be constrained against protein truncating variants (odd ratio ≥ 2.7) (Karczewski et al., 2020)).

Typos:

1. Formatting: Figure 4 sub panels should be renamed to reflect order in text

As suggested, we renamed Figure 4 sub panel labels to reflect order in text.

Reviewer #3

1. The current classification of SVs as coding and non-coding could be clearer. Most larger SVs are likely to affect multiple coding and non-coding regions simultaneously. How does phenoSV handle these cases for prioritization?

Thanks for pointing out the confusion. In this study, we define any SVs affecting one or more coding regions as coding SVs, otherwise as noncoding SVs. We have revised our manuscript shown below to clarify this point (**Materials and Methods**, session 'Training, validation, and test datasets', page 15 paragraph 3).

In Materials and Methods, Training, validation, and test datasets:

We define coding SVs as the ones overlap with at least 1bp on any exons of protein-coding genes according to GENCODE v40 annotations⁶³, otherwise as noncoding SVs. It is important to note that coding SVs include SVs covering coding regions exclusively, as well as those covering both coding and noncoding regions. Conversely, noncoding SVs only cover noncoding regions.

2. Authors claim that phenoSV is unique compared to other methods as it can prioritize all categories of SVs (beyond deletions and duplications). To do this, they convert different categories of SVs into proxy deletion and duplication by applying certain approximations. How do these approximations influence the underlying feature profile of the original SV compared to their proxy? For instance, will the original insertion have a similar feature profile to the transformed proxy deletions?

We thank the reviewer for this comment. One of the unique aspects of PhenoSV is that we attempted to build models beyond simple deletions and duplications. The feature profiles of these SV types can be different from those of proxies. For example, an insertion might inhibit gene expression by disrupting the promoter of this gene, regardless of the exact sequence within the insertion. Thus, the proxy will include the information of regulatory element, but the original feature profile will not. We have added a statement (**Materials and Methods**, session 'PhenoSV for insertions, inversions, and translocations', page 21 paragraph 2) to clarify these differences.

In Materials and Methods, PhenoSV for insertions, inversions, and translocations:

Although PhenoSV was trained using deletions and duplications, it can be adapted to SV types of insertions, inversions, and translocations. We treat deletions and duplications as the basic forms of SVs exerting functional impacts on genes and approximate the impacts of insertions, inversions, and translocations using deletions and duplications (**Figure S1**), *though the original feature profiles of these SV types can be different from those of proxies*. For an insertion, we mainly consider its impacts on disrupting the local genome element by treating it as a 100bp deletion centered at the insertion breakpoint (**Figure S1b**). For an inversion,

3. SVs from certain chromosomes were used as hold-out and others as test sets. What was the rationale for this? Will the model performance remain intact upon randomly assigning SVs in these two groups?

Thank you for this comment which was also raised by another reviewer. We reproduce the response below:

Since structural variants can overlap with each other or influence each other (for example, when residing in the same TAD) in the same chromosome, splitting the dataset into training/validation/hold-out data sets by chromosomes is a commonly used strategy in literature to prevent information leakage from training dataset to validation/hold-out test set. For example, StrVCTVRE used leave-one-chromosome-out for cross validation, and also used chr1, chr3, chr5, and chr7 from ClinVar as the hold-out test set. We certainly acknowledge the existence of genetic architecture differences by chromosomes; here the hypothesis is that if the model trained on specific chromosomes can perform well in other chromosomes, then the model must have extracted useful features that are not dependent on chromosomes. To further assess the validity of the strategy of splitting by chromosomes, as suggested by the reviewer, we also conducted experiments using random splitting of the dataset. As shown in Table S8, random splitting led to improved performance in the hold-out test dataset for both coding SVs (AUC of random split: 0.948; AUC by chromosome: 0.911) and noncoding SVs (AUC of random split: 0.89; AUC by chromosome: 0.86), On the other hand, the performance in the independent test datasets for small SVs (AUC of random split: 0.876; AUC by chromosome: 0.874) and large SVs (AUC of random split: 0.769; AUC by chromosome: 0.770) remained nearly identical for both splitting strategies. Despite these results, it is important to acknowledge that random splitting may lead to inflated performance results within the hold-out test set due to some extent of information leakage. To clarify the rationale of splitting by chromosomes, we revised the manuscript (**Material and Methods**, session ‘**Training, validation, and test datasets**’, page 16 paragraph 2), and we also added results of using the random split strategy in the **supplementary materials** (section ‘**Splitting datasets by chromosomes and splitting datasets by random**’, page 1, paragraph 3).

In Materials and Methods, Training, validation, and test datasets:

We split these SVs, containing only deletions and duplications, into training, validation, and hold-out test datasets based on chromosomes ~~that can produce almost balanced datasets~~ to prevent any information leakage and to ensure the reliability of our test results. The chromosome numbers for these splits were selected to produce balanced datasets concerning pathogenic SVs

and benign SVs. The training set has 14,292 coding SVs (6,609 pathogenic and 7,683 benign) and 330 noncoding SVs (165 pathogenic and 165 benign) from chromosomes 1-10 and 17-22.

4. Why is the non-coding SV training set small? How will it affect the performance of the model?

Despite the existence of a large number of observed noncoding SVs, the majority of them are classified as benign, and only a limited number of them were classified as pathogenic (due to the difficulty of relating noncoding variants to function in the absence of clearcut functional or clinical evidence). This class imbalance can introduce potential biases into our model. To address this issue, we opted to balance the training dataset by matching an equal number of pathogenic and benign noncoding SVs. This step ensures that the model does not solely predict noncoding SVs as benign, which could lead to unreliable model performance. Thus, the number of noncoding SVs in our training dataset is small. Given the challenge of training a noncoding SV-specific model with such a small dataset, we trained the model for both coding SVs and noncoding SVs together. By doing so, the noncoding SVs can benefit from the information learned from coding SVs, resulting in an improved model performance. To further address the reviewer's comments, we have added additional discussions and explanations in the **supplementary materials** (session 'Imbalanced numbers of coding SVs and noncoding SVs for training', page 3, paragraph 2).

5. Why phenoSV performs worse for large coding SVs compared to smaller ones? Also, why genome-wide conservation score is not important for coding SVs?

For the first question, as shown in Figure 3b-c, all models except for CADD-SV yielded lower AUCs in the test set of large SVs. This finding aligns with the results presented in the StrVCTVRE paper, where the authors reported higher AUCs for SVs categorized as either small (<30kb) or large (>500kb) than those with mid-length (30kb-500kb). The model performance decrease for larger SVs could be attributed to potential ascertainment biases between the test dataset of small SVs and the test dataset of large SVs, due to the differences in SV detection technologies. Specifically, larger SVs with sizes over 100kbp are primarily detected by microarrays (with imprecise breakpoints) and are more likely to be reported in literature. On the other hand, smaller SVs, ranging from 50bp to 100kbp, are commonly identified using Next-Generation Sequencing (NGS) techniques; most of these SVs are not reported in literature or documented in databases unless there is clear evidence for pathogenicity. Since previous studies did not specifically discuss this issue, we have now added discussions in the revised manuscript (**Results**, session '**PhenoSV accurately predicts pathogenicity of both coding SVs and noncoding SVs**', page 7 paragraph 1).

For the second question, our results indeed showed that genome-wide conservative features also played an important role in coding SVs (**Figure 3e-f**), but their contribution was relatively less pronounced compared to their impact on noncoding SVs. Since much larger fraction of coding regions tend to be evolutionarily conserved than noncoding regions, it is harder to differentiate pathogenic vs benign coding SVs solely using those conservative features.

In results, PhenoSV accurately predicts pathogenicity of both coding SVs and noncoding SVs:
More importantly, PhenoSV can generate interpretable results by making predictions on the gene level, which is a common limitation for traditional machine-learning methods. Notably, we observed lower AUCs of large SVs than small SVs in the independent test set for all models except CADD-SV (Figure 3b-c). This performance decrease for larger SVs could be attributed to potential ascertainment biases arise from disparities in the techniques used for SV detection (see supplementary materials). Additionally, ...

6. Authors should report precision-recall curves for their model beyond auROC plots.

Thanks for the comment. We added precision-recall curves in Figure S4, and accordingly revised our main manuscript. (Results, session 'PhenoSV accurately predicts pathogenicity of both coding SVs and noncoding SVs', page 6 paragraph 2).

In Results: PhenoSV accurately predicts pathogenicity of both coding SVs and noncoding SVs
As not all methods can produce scores with natural choices of thresholds that distinguish between pathogenic and benign SVs, we used the area under the receiver-operating characteristic curve (AUC) as performance metric to compare different methods. We also reported accuracy, sensitivity, and specificity of PhenoSV in table S3, and area under the precision-recall curves (auPRC) in Figure S4.

7. The number of SVs (beyond deletions and duplications) is very small, particularly for their non-coding SVs. How will this influence the performance of the phenoSV model?

PhenoSV was trained only using deletions and duplications, including coding SVs and noncoding SVs. Since the number of insertions, inversions, and translocations in existing labeled datasets are too small to train a model directly, we only used those SVs as test datasets to evaluate the performance of PhenoSV, when approximating their impacts using deletion/duplication proxies. Thus, the number of insertions, inversions, and translocations will not influence PhenoSV performance. We clarified this point in the revised manuscript (Discussion, page 14 paragraph 2).

In Discussion:

In addition, machine-learning-based models largely rely on existing labeled datasets for training and testing. Since the number of insertions, inversions, and translocations in the existing labeled datasets are too small to train a model directly, PhenoSV used deletions and duplications to approximate the impacts of insertions, inversions, and translocations these SV types to overcome data limitations. Labeled insertions, inversions, and translocations were only used as test datasets to evaluate the performance of PhenoSV and will not influence PhenoSV

performance. However, we should acknowledge that the test dataset of inversions is still small and users should be more cautious when interpreting results on inversions.

8. Authors should comment on the potential issue of model overfitting with such a large amount of feature sets, many of which are likely to be highly correlated.

We acknowledge the reviewer's concern regarding the potential risk of overfitting by using a large feature set, especially if there are strongly correlated features. To address the concern on potential overfitting, we employed several techniques during our analysis. First, we implemented model regularization techniques, including drop out layers and weight decays during model training process. These regularization methods can prevent the model from being too sensitive to the training data and thus can reduce the risk of overfitting. Second, we split our training/validation/hold-out test dataset by chromosomes to ensure the lack of overlaps between SVs across the dataset, and also utilized different independent test datasets for reliable performance evaluations. Third, in the revised manuscript, as suggested by another reviewer, we also introduced a light-weight version of PhenoSV (PhenoSV-light) trained with only 42 important features. Our results suggest that PhenoSV-light exhibited largely comparable prediction accuracy to PhenoSV, except for translocations (Figure S2). Accordingly, we added the discussion on overfitting issues (**supplementary materials**, session '**Potential issues of model overfitting**', page2, paragraph2).

9. The cut-off for classifying pathogenic vs. benign SVs is very lenient, as reported for their transmission analysis. How will a higher threshold for pathogenic SVs influence the overall result?

To address the reviewer's concern, we re-conducted the transmission analysis by assigning the top 30% SVs as pathogenic and the bottom 30% SVs as benign based on PhenoSV score quantiles. (Paternal SVs: ≤ 0.31 as benign, ≥ 0.58 as pathogenic, Maternal SVs: ≤ 0.37 as benign, ≥ 0.71 as pathogenic). Below, we compared the original results with 0.5 as the cut-off value (Table R1) and the new results (Table R2). We found that different thresholds do not influence the overall conclusion of the analysis. Specifically, predicted pathogenic paternal SVs exhibited over-transmission pattern to cases with the transmission rate being 0.71 (0.5 cutoff, binomial test p-value = 0.01) and 0.72 (quantile cutoff, binomial test p-value=0.02), respectively. Predicted benign SVs have transmission rate being 0.64 (0.5 cutoff, binomial test p-value = 0.04) and 0.68 (quantile cutoff, binomial test p-value = 0.07), respectively. Consistent with our original analysis, we observed a slightly larger effect size of over-transmission pattern for paternally inherited pathogenic SVs than benign SVs. Due to the small sample sizes, no statistical significance has achieved when comparing transmission rate between predicted pathogenic SVs and predicted benign SVs (0.5 cutoff: proportion test: p-value=0.656; quantile cutoff: p-value=0.910). The manuscript is revised to reflect these new analysis (**Results**, session '**PhenoSV determines disease-related genes indirectly affected by noncoding SVs**', page 11 paragraph 1). We also added detailed analysis results in **supplementary materials** (session '**Different thresholds of PhenoSVs scores in transmission analysis**', page4, paragraph 3)

In Results: PhenoSV determines disease-related genes indirectly affected by noncoding SVs: After stratifying CRE-SVs into pathogenic ($p_{sv-gene} \geq 0.5$) and benign ($p_{sv-gene} < 0.5$) groups using PhenoSV, we observed the over-transmission pattern of paternally inherited CRE-SVs more evident for predicted pathogenic ones (29/41; transmission rate = 71%; binomial test p-value = 0.01) than predicted benign ones (38/59; transmission rate = 64%; binomial test p-value = 0.04), a slightly larger effect size of over-transmission pattern was observed for paternally inherited pathogenic SVs (29/41; transmission rate = 71%; binomial test p-value = 0.01) than benign SVs (38/59; transmission rate = 64%; binomial test p-value = 0.04). Although statistical significance was not achieved due to limited sample sizes (pathogenic SVs vs benign SVs, proportion test p-value=0.656), these results suggest the values of PhenoSV in determining pathogenic genes indirectly affected by noncoding SVs. When classifying pathogenic CRE-SVs and benign CRE-SVs, different thresholds of $p_{sv-gene}$ (such as top and bottom 30% quantiles) can be used and we found that different thresholds do not influence the overall conclusion of the analysis (supplementary materials, Table S9-S10).

Table R1. Transmission analysis with pathogenicity threshold of 0.5.

	PhenoSV stratification	paternal total	father transmitted	father untransmitted	father transmitted rate	father p	maternal total	mother transmitted	mother untransmitted	mother transmitted rate	mother p
case	all	100	67	33	0.67 (0.57, 0.76)	0.0008737	79	47	32	0.59 (0.48, 0.70)	0.1147
	predicted pathogenic	41	29	12	0.71 (0.54, 0.84)	0.01151	47	28	19	0.60 (0.44, 0.74)	0.243
	predicted benign	59	38	21	0.64 (0.51, 0.76)	0.03634	32	19	13	0.59 (0.41, 0.76)	0.3771
control	all	26	16	10	0.62 (0.41, 0.80)	0.3269	17	10	7	0.59 (0.33, 0.82)	0.6291
	predicted pathogenic	11	7	4	0.64 (0.31, 0.89)	0.5488	13	8	5	0.62 (0.32, 0.86)	0.5811
	predicted benign	15	9	6	0.60 (0.32, 0.84)	0.6072	4	2	2	0.50 (0.07, 0.93)	1

Table R2. Transmission analysis with pathogenicity threshold of 30% and 70% quantiles.

	PhenoSV stratification	paternal total	father transmitted	father untransmitted	father transmitted rate	father p	maternal total	mother transmitted	mother untransmitted	mother transmitted rate	mother p
case	all	100	67	33	0.67 (0.57, 0.76)	0.0008737	79	47	32	0.59 (0.48, 0.70)	0.1147
	predicted pathogenic	29	21	8	0.72 (0.53, 0.87)	0.02412	24	11	13	0.46 (0.25, 0.67)	0.8388
	predicted benign	31	21	10	0.68 (0.49, 0.83)	0.07076	25	12	13	0.48 (0.28, 0.69)	1
control	all	26	16	10	0.62 (0.41, 0.80)	0.3269	17	10	7	0.59 (0.33, 0.82)	0.6291
	predicted pathogenic	9	6	3	0.67 (0.30, 0.93)	0.5488	5	2	3	0.4 (0.05, 0.85)	1
	predicted benign	7	4	3	0.57 (0.18, 0.90)	1	4	2	2	0.5 (0.07, 0.93)	1

10. The current phenoSV approach doesn't consider tissue-specificity for non-coding SVs. As discussed by the authors, this is one of the biggest drawbacks of phenoSV. Why can't authors utilize tissue-specific annotation or epigenomic signals in the current phenoSV framework?

Thank you for the insightful comments on tissue-specificity of non-coding SVs. To develop a tissue-specific model for SVs, we need to annotate SVs using tissue-specific expression profiles and epigenomics features. We did not pursue this approach for the following reasons: First, it would require a significantly larger number of labeled SVs compared to the current dataset, to achieve accurate tissue-specific predictions. Such extensive labeled data for each tissue is currently not available. Second, a considerable number of labeled SVs in the dataset lack tissue or disease information, which further restricts the availability of data that could be used for training. Third, not all disease-relevant tissues have experimental mapping data, and in some cases, the quality of available data may be suboptimal, adding noises to the model. Given these considerations, we trained the PhenoSV model with aggregated annotation and epigenomic features but then use disease/phenotype information to further refine the predictions.

As a step towards addressing this limitation, the current PhenoSV framework allows users to incorporate tissue specific TAD information to define candidate gene sets during inference. This enhancement enables users to leverage tissue-specific context for certain analyses. To facilitate this process, we have updated the user manual of the PhenoSV GitHub repository, providing detailed guidance on how to incorporate user-defined TAD annotations. This update is also described in the revised manuscript (**Discussion**, page 14 paragraph 2).

In Discussion:

Since not all labeled SVs have the corresponding tissue source information and TAD annotations are tissue-dependent, we used a sub-optimal distance-based strategy to determine the candidate gene sets for all SVs during training. Nevertheless, tissue-specific TAD annotations can be used to derive more defined candidate gene sets when using PhenoSV. This capability is facilitated by the current command-line tool of PhenoSV, which enables users to employ their own TAD annotations, such as tissue-specific TAD, for specific analyses. As more experimental data, like Hi-C data, becomes available and expand existing tissue-specific genome annotations, more efficient approaches will be explored for further improvements. In addition,

Reviewer #1 (Remarks to the Author):

The authors have addressed all of my concerns. No further suggestions.

Reviewer #2 (Remarks to the Author):

The authors have very thoughtfully and seriously addressed my comments as well as the comments of the other reviewers. Although no method or model can be perfect, I am satisfied that the manuscript and accompanying model will be useful for the field.

Reviewer #3 (Remarks to the Author):

The authors have addressed the majority of the concerns. However, their current framework has significant limitations and potential applicability for identifying pathogenic SVs that drive disease by affecting non-coding regions on the genome. This drawback is primarily due to the limited training sample size and lack of tissue-specific epigenomic/ functional genomics data for training their non-coding SV model. A clear acknowledgment/elaboration of this issue in the discussion section is warranted.

Response to Comments

We thank again for reviewers' generous and constructive comments on our manuscript titled "PhenoSV: Interpretable phenotype-aware model for the prioritization of genes affected by structural variants". We have fully addressed these concerns. The changes that we made in the main manuscript were highlighted in **red fonts**.

Reviewer #1 (Remarks to the Author):

The authors have addressed all of my concerns. No further suggestions.

Reviewer #2 (Remarks to the Author):

The authors have very thoughtfully and seriously addressed my comments as well as the comments of the other reviewers. Although no method or model can be perfect, I am satisfied that the manuscript and accompanying model will be useful for the field.

Reviewer #3 (Remarks to the Author):

The authors have addressed the majority of the concerns. However, their current framework has significant limitations and potential applicability for identifying pathogenic SVs that drive disease by affecting non-coding regions on the genome. This drawback is primarily due to the limited training sample size and lack of tissue-specific epigenomic/ functional genomics data for training their non-coding SV model. A clear acknowledgment/elaboration of this issue in the discussion section is warranted.

According to the reviewer's suggestion, we further addressed the limitation in the Discussion section (page 14, paragraph 2).

Similarly, the number of noncoding SVs is limited in our training dataset. To train a model that can accurately predict the pathogenicity of noncoding SVs, we devised a strategy that makes the input features of coding SVs and noncoding SVs "look alike" (see Supplementary Materials). This approach enabled us to utilize information from coding SVs to enhance the training for noncoding SVs. Yet, the scarcity of pathogenic noncoding SVs in our training and testing dataset, along with the lack of tissue-specific functional annotations, remains a notable limitation that requires further improvement once appropriate datasets become available.